# Do Depth-Grown Models Overcome the Curse of Depth? An In-Depth Analysis

## Abstract

Gradually growing the depth of Transformers during training can not only re-
duce training cost but also lead to improved reasoning performance, as shown by
MIDAS (Saunshi et al., 2024). Thus far, however, a mechanistic understanding of
these gains has been missing. In this work, we establish a connection to recent
work showing that layers in the second half of non-grown, pre-layernorm Trans-
formers contribute much less to the final output distribution than those in the first
half—also known as the *Curse of Depth* (Sun et al., 2025b; Csordás et al., 2025).
Using depth-wise analyses, we demonstrate that growth via gradual middle stack-
ing yields more effective utilization of model depth, alters the residual stream
structure, and facilitates the formation of permutable computational blocks. In
addition, we propose a lightweight modification of MIDAS that yields further im-
provements in downstream reasoning benchmarks. Overall, this work highlights
how the gradual growth of model depth can lead to the formation of distinct com-
putational circuits and overcome the limited depth utilization seen in standard
non-grown models.

## 1 Introduction

The remarkable success of large language models (LLMs) has been accompanied by immense com-
putational and energy demands. This trend of training larger and larger networks is correlated with
the increasing depth of model architectures (Kaplan et al., 2020; Hoffmann et al., 2022). As Trans-
formers (Vaswani et al., 2017) lack recurrence, their computational capacity is directly linked to
their depth. Greater depth enables more complex computations and improves capabilities like rea-
soning, compositional generalization and goal reaching (Petty et al., 2023; Lad et al., 2024; Wang
et al., 2025). However, this pursuit of greater scale uncovers a critical inefficiency, as training such
models is extremely resource-intensive (Varoquaux et al., 2025).

A core issue of the current paradigm is the observation that not all layers contribute equally to the
final model's performance (Yin et al., 2023; Gromov et al., 2024; Li et al., 2024; Men et al., 2024).
Csordás et al. (2025) and Sun et al. (2025b) demonstrate that deeper layers of modern pre-layer
Transformers tend to be less effective than their earlier counterparts, with many layers in the second
half of the model contributing minimally to the final output—also known as the *Curse of Depth* (Sun
et al., 2025b). This observation, which highlights a kind of over-parametrization, is supported by
findings that various architectures are remarkably robust to perturbations like skipping layers without
significant performance loss (Lad et al., 2024; Yin et al., 2023). The Curse of Depth represents a
major resource inefficiency in today's paradigm. As highlighted by Csordás et al. (2025), addressing
these limitations is a pressing need for the community to avoid wasting valuable resources and to
develop more efficient architectures that can leverage deep layers effectively.

A promising solution lies in gradually grown architectures, which dynamically expand a model's
depth or width during training. These novel training strategies, such as gradual stacking (Gong et al.,
2019; Reddi et al., 2023), enable efficient training by using layers from a smaller model to initialize
the next stage. Of particular interest is the `MIDAS` method (Saunshi et al., 2024), which gradually
increases depth by inserting new layers into the middle of the model. `MIDAS` has been shown
not only to speed up training but also to improve performance on reasoning-heavy benchmarks,
suggesting that this growth procedure introduces a favourable inductive bias. However, a clear
mechanistic understanding of these gains has so far been missing.

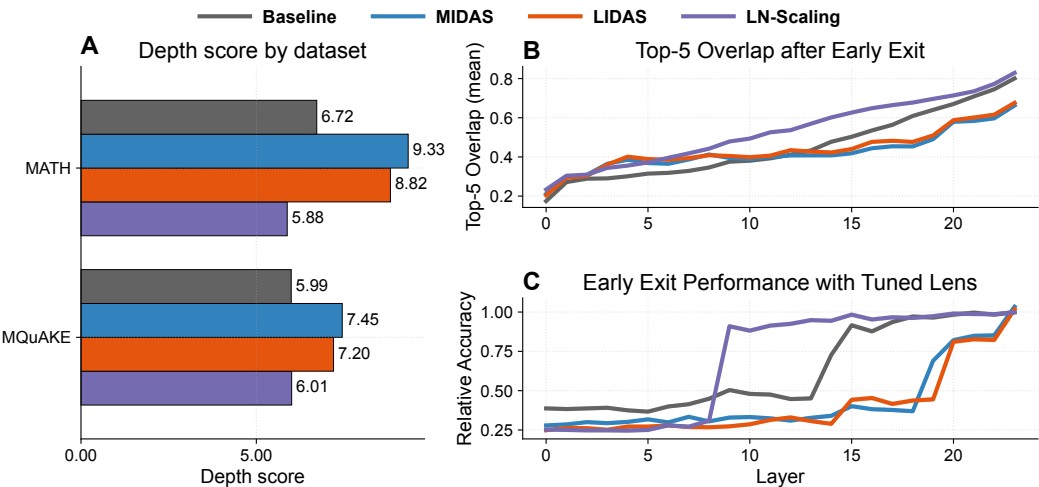

**Figure 1: Depth-grown models use their depth more (1.7B).** (A) Depth score (Csordás et al., 2025) on MATH (Hendrycks et al., 2021) and MQuAKE (Zhong et al., 2023). Grown models (MIDAS, LIDAS) have consistently higher depth scores. (B) Top-5 overlap between each layer's early-exit vocabulary and model's final vocabulary on 20 prompts from GSM8K (Cobbe et al., 2021). Both grown models studied in this work exhibit lower overlap at later layers, indicating that these later layers still contribute additional features necessary for the final prediction. (C) Early-exit relative accuracy versus layer on *Variable Assignment Math* reasoning primitive. The baseline reaches near its final performance early, whereas accuracy for MIDAS and LIDAS continues to rise up to the last layer. Using these metrics, however, LN-Scaling shows no discernible benefit over the baseline in depth utilisation.

In this work, we establish a direct connection between gradual depth growth and the "Curse of Depth" (Sun et al., 2025b), providing a mechanistic understanding of how gradual depth growth procedures can lead to more effective utilisation of a model's depth by altering its computational structure. Using analysis tools from Belrose et al. (2023) and Csordás et al. (2025), we show that gradual stacking counteracts the patterns of diminishing returns observed in non-grown models and gives rise to qualitatively different depth-wise computation. We summarize our contributions below:

- **MIDAS reproduction on different backbones.** We reproduce the core MIDAS results on SmolLM-v1 backbones (360M and 1.7B), trained with autoregressive next-token prediction, confirming that gradual depth growth improves reasoning performance over a conventionally trained, non-grown baseline with a 1.29x improvement in training speed.

- **Novel gradual depth growth strategy LIDAS.** Based on the motivation of MIDAS, i.e., its connection to Looped Transformers and symmetric functional behaviour, we propose LIDAS, an improved growing strategy that duplicates the layer-wise middle rather than the block-wise middle while preserving the inductive bias of growing. Across scales, LIDAS matches or exceeds MIDAS and conventionally trained models in reasoning benchmarks without degrading Negative Log-Likelihood (NLL) or knowledge performance. Additionally, LIDAS results in a more symmetric weight structure than MIDAS and aligns its attention sublayers in the middle of the network better with the residual stream.

- **Extensive analysis of improved and altered depth utilization in grown models.** We provide an in-depth analysis of how gradual depth growth alters computation and representation in LLMs, providing mechanistic insights into these findings. We show that grown models utilize their depth more efficiently than conventionally trained baselines, by reaching their final performance only at the very last layer. Furthermore, we demonstrate that grown models develop permutable computational blocks in the middle of the network, with each layer within a block fulfilling a specific cyclical role.

Overall, this work provides a first mechanistic understanding of how gradual depth growth can counteract the Curse of Depth, potentially leading to more efficient and capable language models.

## 2 RELATED WORK

**Growing Neural Networks.** Early on, researchers recognized the advantages of training neural networks one layer at a time (Hinton et al., 2006; Bengio et al., 2006) to overcome the challenges of learning long-term dependencies with gradient descent (Bengio et al., 1994). More recently, this concept has been re-explored for large language models (LLMs) through gradual stacking (Gong et al., 2019; Reddi et al., 2023; Du et al., 2024) and depth up-scaling (Kim et al., 2023). Alternative growing strategies include masked structural growth (Yao et al., 2023), function-preserving expansions (Gesmundo & Maile, 2023) and learned linear growth operators (Wang et al., 2023).

**Adaptive Architectures.** A potentially complementary strategy to growing is using adaptive architectures that dynamically adjust their computational graph or parameters based on their input data by using a larger, pre-trained network more efficiently, including mixture of experts approaches (Jacobs et al., 1991; Shazeer et al., 2017; Csordás et al., 2024) or early exiting (Teerapittayanon et al., 2016; Xin et al., 2020). Recent approaches apply adaptive token-level computations (Bae et al., 2025), nested models for elastic inference (Devvrit et al., 2024) or depth-wise looping (Giannou et al., 2023; Yang et al., 2023; von Oswald et al., 2025).

**Depth of Neural Network Architectures.** While depth is a key factor correlated with network performance (Csordás et al., 2025), recent research has found that deeper layers in LLMs are often redundant and less effective. (Sun et al., 2025b) have termed this phenomenon the Curse of Depth, which suggests deeper layers contribute minimally to learning. To counteract, they propose LN-Scaling, which scales each layer's LayerNorm activation to suppress depth-induced variance growth so deeper layers learn usefully. Complementary to such architectural and normalisation changes, Dey et al. (2025) introduce CompleteP, a depth-and width-aware parameterisation that achieves depth-wise hyperparameter transfer, enabling compute-efficient training of very deep Transformers across a broad range of width–depth aspect ratios.. Studies on models like GPT-2 show that their middle and deep layers exhibit remarkable robustness to significant perturbations, including layer swapping and deletion (Yin et al., 2023; Lad et al., 2024). This over-provisioning has inspired various layer intervention strategies, such as skipping, swapping, or parallelization, to improve efficiency (Lad et al., 2024; Sun et al., 2025a).

**Reasoning.** For solving challenging tasks, recent work has shifted focus to recurrence and looping (Geiping et al., 2025; Saunshi et al., 2025) to improve model reasoning and leverage depth scaling for enhanced internal "thinking" (Chen et al., 2025). These methods scale up test-time computation to allow models to iteratively refine their answers. Complementary to these approaches, our work focuses on identifying and leveraging computational blocks within depth-grown neural networks to improve reasoning, rather than relying on a fixed, recurrent process.

## 3 TWO DEPTH-GROWN TRANSFORMERS: MIDAS & LIDAS

In this section, we first formalise the growth operator on a fixed architecture class $\mathcal{F}$ and recover MIDAS (Saunshi et al., 2024) as a special case. We then introduce LIDAS, which inserts a new middle block constructed by interleaving its neighbours to provide a stronger initialisation. Finally, using models from the SmolLM-v1 family (Ben Allal et al., 2024), we present empirical results on aggregated reasoning and knowledge benchmarks, showing that both gradual-depth growing methods outperform a conventionally trained, non-grown baseline and LayerNorm-Scaling (LN-Scaling Sun et al. (2025b)) on reasoning tasks while remaining on par with general language-modelling performance.

### 3.1 THE GROWING OPERATOR

We fix a base architecture class $\mathcal{F}$ (width, heads, embedding size, etc. are fixed) and vary only depth. Let $f_L \in \mathcal{F}$ denote a model with $L$ Transformer layers, written as an ordered list $f_L = [\ell_0, \ldots, \ell_{L-1}]$. A (depth) *growth operator* $G : \mathcal{F} \times \mathbb{N} \to \mathcal{F}$ maps an $L$-layer model to an $(L + b)$-layer model, such that $G(f_L; b) = f_{L+b}$, where $b \in \mathbb{N}$ is the *block size* (the number of layers added per growth step). Following Saunshi et al. (2024), we consider growth operators that insert new layers in the centre of the model and keep the block size $b$ fixed across growing stages.

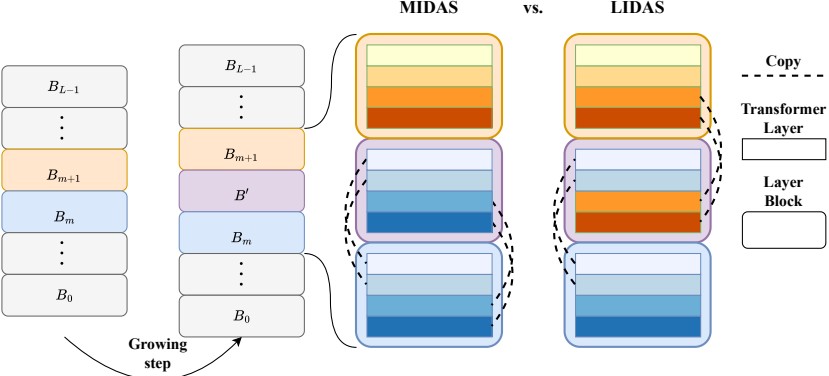

**Figure 2: Illustration of growing strategies with block size 4**: MIDAS vs. LIDAS, with an even number of existing blocks. MIDAS (Saunshi et al., 2024) simply copies $B' = B_m$, which is the block preceding mid-depth. When seen from a block-wise perspective instead of a layer-wise perspective, our proposed variant LIDAS may be interpreted as forming $B'$ from the two blocks surrounding the mid-depth by combining the first two layers of $B_{m+1}$ with the last two layers of $B_m$. This small difference in initialization leads to significantly improved performance as shown in Table 1.

The following strategies use layer duplication to initialize new layers within the newly inserted block. This consists of deep copying all parameters within the layer, including their optimizer state. The result is two initially identical copies at different depths of the model, thus allowed to diverge as training continues.

**MIDAS.** When depth increases by $b$ layers per stage, at stage $n$ we have $L = nb$ and can partition $f_L$ into $n$ contiguous blocks of size $b$ :

$$f_L = [B_0 \,\|\, B_1 \,\|\, \cdots \,\|\, B_{n-1}], \quad B_j = [\ell_{jb}, \ldots, \ell_{(j+1)b-1}]. \tag{1}$$

Let $m_b = \lceil \frac{n}{2} \rceil - 1$ denote the *middle block* index. Middle gradual stacking inserts a new block $B'$ immediately after $B_{m_b}$, i.e.

$$G(f_L; b) = [B_0 \,\|\, \cdots \,\|\, B_{m_b} \,\|\, B' \,\|\, B_{m_b+1} \,\|\, \cdots \,\|\, B_{n-1}]. \tag{2}$$

If $B' = B_{m_b}$ (copying the middle block), we recover MIDAS as proposed in Saunshi et al. (2024).

**LIDAS.** Since we are constrained by the block patterning in MIDAS, we propose Layer-wise mIDdle grAdual Stacking, or LIDAS, in which we consider the *middle layer* $m_l = \lceil \frac{L}{2} \rceil$ to be the central point of the growing operation. We then construct a new block $B' = [l_{m_l - \lceil b/2 \rceil}, \ldots, l_{m_l + \lfloor b/2 \rfloor}]$, around the middle layer $l_{m_l}$, which is inserted after the layer $l_{m_l + \lfloor b/2 \rfloor}$. For an odd number of blocks, MIDAS and LIDAS coincide by selecting the same layers. They differ for an even number of blocks, shown from a block-wise perspective in Fig. 2. Further details can be found in Appendix A.

**Training runs and schedules.** A training run is specified by (i) the model class $\mathcal{F}$, (ii) the target depth $L_{\text{final}}$, (iii) the initial depth $L_0$ (typically $L_0 = b$), (iv) a fixed block size $b$, and (v) a stage schedule $\{T_s\}_{s=0}^{S-1}$ (training steps per stage). Starting from $f_{L_0}$, after each stage $s$, we apply $G(\cdot; b)$ to obtain $f_{L_{s+1}}$ with depth $L_{s+1} = L_s + b$. We repeat until $L_{S-1} = L_{\text{final}}$.

## 3.2 EXPERIMENTS

**Setup.** We evaluate and compare the two growing methods, MIDAS and LIDAS, against a conventionally non-grown baseline and one alternative method LN-Scaling. We use the 360M and 1.7B models from the SmolLM-v1 family (Ben Allal et al., 2024), trained on 200B and 400B tokens respectively, to probe scaling behaviour. All models are trained from scratch on the SmolLM-Corpus, a curated mixture of educational and synthetic texts as well as mathematics and code. Due to their favourable efficiency–performance trade-off, these models enable competitive evaluation within a constrained computational budget. For all grown models we present, we use the block size $b = 4$ and a PROP-1 growing schedule (see Appendix A for details).

| | | Standard cooldown | | | | | | Math cooldown | |
|---|---|---|---|---|---|---|---|---|---|
| | | **Open-book Q&A** (F1 ↑) | **Closed-book Q&A** (F1 ↑) | **Lambada** (Acc ↑) | **Hellaswag** (Acc ↑) | **MathWorld** (Acc ↑) | **Primitives** (Acc ↑) | **MathWorld** (Acc ↑) | **Primitives** (Acc ↑) |
| | **Holdout Set** (NLL ↓) | | | | | | | | |
| **360M** Baseline | 2.18 | 22.89 | 14.50 | 43.35 | 39.97 | 3.69 | 30.06 | 8.10 | 33.12 |
| LN-Scaling | **2.16** | 23.14 | **14.89** | 42.17 | 40.0 | 2.89 | **31.38** | 8.45 | 41.26 |
| MIDAS | 2.18 | 24.57 | 13.75 | 43.31 | 40.36 | **4.39** | 28.18 | **13.43** | 35.14 |
| LIDAS | **2.16** | **26.63** | 14.57 | **44.03** | **40.58** | 4.36 | 31.20 | 12.30 | **50.36** |
| **1.7B** Baseline | **1.96** | 29.57 | 18.61 | 50.05 | 46.28 | 13.75 | 34.84 | 23.28 | 42.77 |
| LN-Scaling | 1.97 | 29.11 | 18.63 | 48.94 | 45.44 | 11.0 | 44.38 | 17.84 | 50.58 |
| MIDAS | 1.97 | 28.80 | 18.50 | 50.81 | 46.19 | 16.07 | 40.88 | 24.01 | **55.46** |
| LIDAS | **1.96** | **29.84** | **19.08** | **51.41** | **46.32** | **18.59** | **47.34** | **24.60** | 53.00 |

**Table 1: Performance comparison of a standard transformer baseline, LayerNorm-Scaling, and the two grown models `MIDAS` and `LIDAS`.** We reproduce the findings of Saunshi et al. (2024) and observe that grown models match the baseline in training objective (NLL), standard Q&A benchmarks as well as Lambada. Grown models, especially `LIDAS`, outperform the non-grown baseline on reasoning-heavy tasks such as MathWorld and Primitives. `LN-Scaling` on the other hand, achieves only minor improvements, which diminish when scaling to the larger model.

**Benchmarks.** We report negative log-likelihood (NLL) on a held-out validation set from the SmolLM-Corpus. We follow the knowledge and reasoning benchmarking suite reported in Saunshi et al. (2024). The knowledge-based benchmarks are split into Open-book Q&A with provided context (TyDiQA-GoldP, SQuADv2, DROP, QuAC, CoQA), and Closed-book Q&A without context (TriviaQA, TyDiQA-NoContext, NaturalQuestions, WebQuestions), evaluated zero-shot. We additionally add Lambada (Paperno et al., 2016) and HellaSwag (Zellers et al., 2019) in their classical settings. For reasoning, we report the aggregated performance on MathWorld problems (SVAMP (Patel et al., 2021), ASDiv (Miao et al., 2021), and MAWPS (Koncel-Kedziorski et al., 2016)) and reasoning primitives, which are a suite of synthetic tasks designed by Saunshi et al. (2024) to specifically investigate reasoning performance on a smaller scale, both evaluated under five-shot prompting as done previously. For the exact score breakdown, we refer to Appendix C.

**Results.** Aggregated results are shown in Table 1. Consistent with the findings of Saunshi et al. (2024), we observe that depth-grown models (both `MIDAS` and `LIDAS`) outperform the baseline on reasoning-heavy tasks (i.e., MathWorld and Reasoning Primitives). On the remaining benchmarks (Open-book Q&A, Closed-book Q&A, and Lambada), we observe little deviation from the baseline model with `LIDAS` being slightly superior to `MIDAS`. In addition, we observe a 29% speedup in training for depth-grown models over a conventionally trained, non-grown baseline (Table 7). `LN-Scaling`, on the other hand, largely tracks the baseline on NLL and knowledge tasks, offering only small gains at 360M that disappear at 1.7B, while trailing the grown variants on reasoning. In summary, our results reproduce the observation of Saunshi et al. (2024). `MIDAS` outperforms a conventionally trained baseline on reasoning-heavy tasks, and our proposed method `LIDAS` further strengthens this effect without degrading NLL at 1.7B. To stabilise results on MathWorld, we additionally report the performance of models which are finetuned on the OpenWebMath dataset. While the relative order stays the same, we observe for the 360M model that the improvements for the grown models become more pronounced. However, the reasons behind these gains remain unclear. Therefore, we turn next to a detailed analysis of the 1.7B models, aiming to characterize how `MIDAS` and `LIDAS` may mechanistically differ from the baseline and how this could lead to improved performance in reasoning tasks.

## 4 DEPTH ANALYSIS

Motivated by the confirmed observations that gradually depth-grown Transformers seem to yield increased reasoning abilities, we investigate here how gradual depth growth reshapes computation across depth in fully trained models. To this end, we first examine early-exiting performance for every layer with TunedLens (Belrose et al., 2023) to test how much performance degrades when we exit early. Next, we run various interventions on the models, such as swapping contiguous blocks of layers, to test whether grown models form permutable circuits, and how sensitive each method is to late-layer ablations. We then analyse the layer-wise roles within blocks by measuring the similarity between each layer's contribution and the residual stream. Finally, we compare `MIDAS` with `LIDAS` on weight symmetry and contribution per attention matrix, connecting it to benchmark results of the

previous section. All analyses are conducted on the 1.7B variant described in Table 1, and analogous results for the 360M models are reported in Appendix D.2. A detailed description of the setups for each analysis can be found in Appendix B. For notation, we follow Csordás et al. (2025): $h_{i+1}$ denotes the residual stream after transformer layer $l_i$, $a_i$ the layer's attention output and $m_i$ the output of the MLP.

## 4.1 DOES DEPTH GROWTH LEAD TO DIFFERENT DEPTH UTILIZATION?

**Hypothesis.** Gradual depth-grown Transformers (with MIDAS and LIDAS) utilize model depth more efficiently than conventionally trained, non-grown Transformer baselines.

**Evidence.** Skipping late layers degrades prediction accuracy substantially more for MIDAS and LIDAS than for the baseline, which coincides with an increased depth score.

**Experiments.** To investigate the contribution of deeper layers, we evaluate intermediate representations via a Tuned Lens (Belrose et al., 2023). Concretely, for each layer $l_i$, we train a small affine adapter on a split of FineWeb-Edu (Penedo et al., 2024) that maps that layer's residual output to the hidden representation consumed by the final normalization; we then obtain logits by applying the model's final normalization and unembedding (Belrose et al., 2023), enabling early-exit at every depth[1]. Subsequently, we quantify depth utilization by the top-5 vocabulary overlap of their predicted vocabularies (Fig. 1B), and early-exiting accuracy on a reasoning primitive (Fig. 1C). Finally, we compute the depth score (Csordás et al., 2025) to summarize where computation occurs along the network by estimating each layer's influence on future tokens (Fig. 1A). For further details, we refer to Appendix B, and for results on the 360M model, to Appendix D.2.

**Interpretation.** For MIDAS and LIDAS, Fig. 1B shows that early-exit predictions differ substantially more from the final logits than in the baseline (lower top-5 overlap), indicating that later layers in the grown models add features to the residual stream that are required for the final prediction. In Fig. 1C, the baseline reaches its final performance by Layer 18, whereas accuracy for both grown models continues to improve up to the last layer. Lastly, Fig. 1A consistently reports higher depth scores for the grown models across datasets, most notably on math tasks, indicating that more computation is concentrated in later layers. Interestingly, LN-Scaling, designed by Sun et al. (2025b) to reduce output variance in deeper layers of pre-layernorm transformers and thereby improve the utilization of later layers, exhibits the opposite behavior here. Concretely, LN-Scaling shows a lower depth score, stronger overlap with final tokens in earlier layers, and achieves peak performance earlier than both the baseline and grown models for the 1.7B (Fig. 1) and 360M (Fig. 14) models.

## 4.2 DOES DEPTH GROWTH FORM PERMUTABLE COMPUTATIONAL BLOCKS?

**Hypothesis.** Non-grown models depend on their specific layer ordering. Depth-grown models, on the other hand, develop computational blocks that are robust to block-level ordering interventions.

**Evidence.** Reduced performance degradation under multi-layer perturbations indicates lower layer order dependence and greater robustness of MIDAS and LIDAS.

**Experiments.** To evaluate layer functional independence, we swap contiguous sub-blocks of sizes $\{1, 2, 4, 8\}$ and measure the effect on downstream performance. While these experiments can indicate robustness, we can also observe how commutative sub-blocks are, as the local order of layers is preserved when swapping larger blocks.

**Interpretation.** Swapping just single layers does not affect the performance of the baseline and grown models much (Fig. 3), except for the input layers. This observation aligns with findings from Lad et al. (2024). If we increase the number of consecutive layers that we swap, the accuracy of the baseline quickly starts to deteriorate. In contrast, grown models allow swapping blocks of up

---

[1]Note that this should result in more accurate predictions than naively applying the unembedding matrix at every layer (LogitLens (Nostalgebraist, 2020)), as done in Csordás et al. (2025).

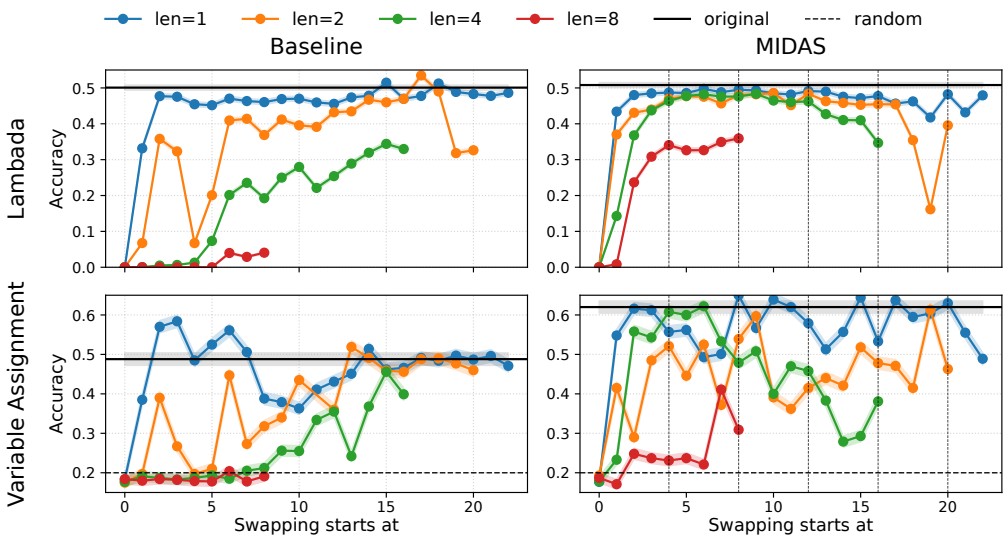

**Figure 3: Effect of swapping blocks of layers on Lambada (top row) and the reasoning primitive *Variable Assignment Math* (bottom row).** MIDAS is more robust to interventions for larger blocks in the middle of the network: the degradation in performance for MIDAS is much smaller for swapping blocks of larger sizes {2, 4, 8} compared to the baseline, especially for Lambada. In Appendix Fig. 9, we present results including LIDAS.

to size four with a relatively small decrease in performance, and we observe even less performance degradation when swapping blocks, indicating less order dependence of these blocks. The grown models even reach non-random performance when swapping middle 8-layer blocks compared to the baseline, whose performance drops to random. In general, the degradation is lower on the language-modelling task (Fig. 3 top row) compared to the reasoning primitive (bottom row). Taken together, these effects are most consistent with the emergence of computational blocks whose internal order matters less than the presence of the block as a unit, matching the qualitative behaviour in Fig. 3.

### 4.3 DOES GRADUAL GROWTH FORM LAYER-WISE PATTERNS?

> **Hypothesis.** The block-wise growing introduces a cyclical pattern in the architecture such that each layer within a block fulfils a certain role.
>
> **Evidence.** The contribution of the attention sublayer, in norm and cosine similarity, repeats in each block. When performing causal interventions, the effect for each layer within a block also repeats. Reversing the order of layers within and especially across blocks destroys the performance of grown models more than swapping, where local order is more preserved.

**Experiments.** Using the tools of Csordás et al. (2025), we compute for each attention layer its cosine similarity to the residual stream ($\frac{a_i \cdot h_i}{||a_i||||h_i||}$) and its mean relative contribution $\frac{||a_i||}{||h_i||}$. We then intervene by skipping a transformer layer or sublayer and track the relative changes in downstream computations under two regimes: (i) propagated: zeroing that component's contribution to the residual stream and forwarding this change to all downstream layers; and (ii) local: removing a layer's contribution from all subsequent inputs separately to isolate pairwise source–target dependencies. Finally, we assess the effect of reversing the order of four consecutive layers and comparing the outcome to results from Fig. 3. A detailed explanation of the interventions can be found in Appendix B.

**Interpretation.** Grown models exhibit a highly cyclical pattern in the middle, where the effect is especially visible for the attention sublayer (Fig. 4). The mean relative contribution of the attention sublayer always grows from its lowest point at the first layer of every block to its highest point at the last layer of the block. The highest spike across depth is always at the final layer of the last block in

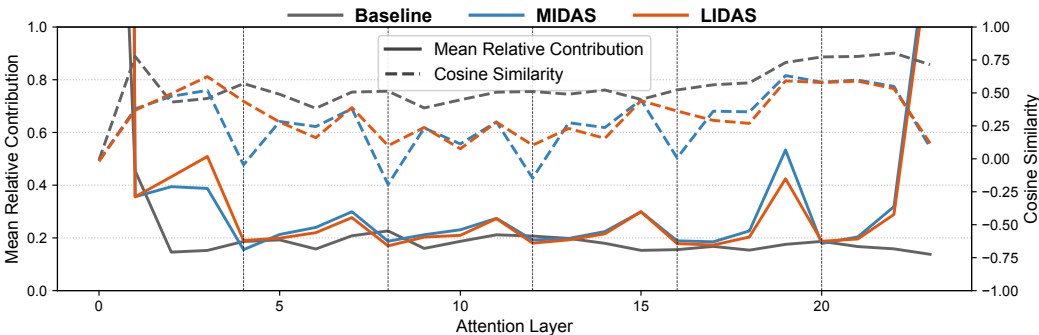

**Figure 4: Attention layer contributions to the residual stream.** Grown models exhibit a highly cyclical pattern in the centre of the network. The mean relative contribution of the attention sublayer to the residual stream increases throughout a block (whose first layer is denoted by a vertical line) and has its largest contribution in the last layer of each block. While the cosine similarity between the output of each attention layer and the residual stream is relatively flat for the baseline, the pattern for the grown models again depends on the block size and the relative position of the layers within each block. Notably in MIDAS, the first attention sublayer of a block has a very low cosine similarity to the residual stream, while for LIDAS the attention contributions align more with the residual stream.

the middle of the network, i.e., the overall second-to-last block. For MIDAS the cosine similarity of the attention sublayers in the middle, similarly to their contributions, always rises from around zero, adding orthogonal features, or slightly negative, weakening or erasing features, to the highest but only slightly positive cosine similarity at the end of each block. The pattern for LIDAS is a little bit less clear, but the cosine similarity never drops as low as MIDAS, potentially adding features from subspaces that are better aligned with the residual stream across the whole block.

Turning towards interventions, by skipping a layer, the most pronounced disruption to future computations arises when skipping the second layer of each block (aside from the earliest layers), with often the biggest observed relative change in the immediate layer after it, i.e., in each block's third layer (Fig. 5a). We hypothesize that the second layer prepares features for future computations. If we measure the relative change on the following layers directly, we notice a clear and striking pattern (Fig. 5b). For future computations, the third layer of every block directly depends on the features of almost all previous layers, potentially performing an aggregating operation. The direct change of removing the output of a previous layer is less on deeper blocks that can depend on more inputs simultaneously, i.e., visually a fading pattern. The last block mostly depends on the final aggregation and strengthening of relevant features performed by the second-to-last block.

Reversing the order of four consecutive layers (Fig. 6) reduces performance in the grown model far more than swapping pairs of two or four layers (len = 2, 4 in Fig. 3), where local order is more preserved. The baseline is comparatively robust to reversals involving later layers, which aligns with the hypothesis from Csordás et al. (2025) that later layers in pre-layernorm transformers refine the current output distribution with less order dependence. By contrast, the grown model is most brittle when the reversal straddles block boundaries (last two layers and first two layers of consecutive blocks), showcasing that the order of layers within a block matters.

### 4.4 DOES GROWING STRATEGY LEAD TO DISTINCT BEHAVIOUR?

**Hypothesis.** Compared to MIDAS, LIDAS produces more symmetric weights and utilises its depth more effectively, making better use of central layers towards better empirical performance.

**Evidence.** In LIDAS, inter-block cosine similarities are higher and more symmetric about the centre. Skipping the first attention sublayer in the middle blocks causes larger relative changes in the hidden state of the token under consideration.

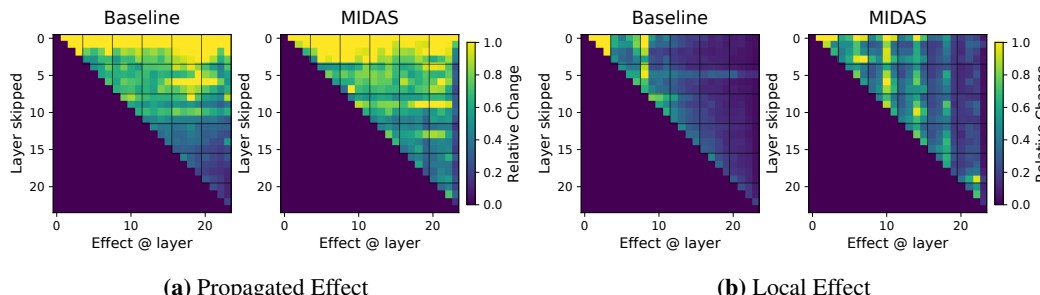

**(a)** Propagated Effect        **(b)** Local Effect

**Figure 5: Baseline vs. MIDAS. Effect of skipping a layer on downstream layer contributions for *future* tokens**. (a) MIDAS relies more on later layers than the baseline for future computations. Especially skipping the second layer of each mid-block strongly impacts the immediately following layer. (b) For MIDAS, the third layer of every block in the middle directly depends on all previous computations. We refer to Fig. 12 and Fig. 13 in the Appendix for results including LIDAS.

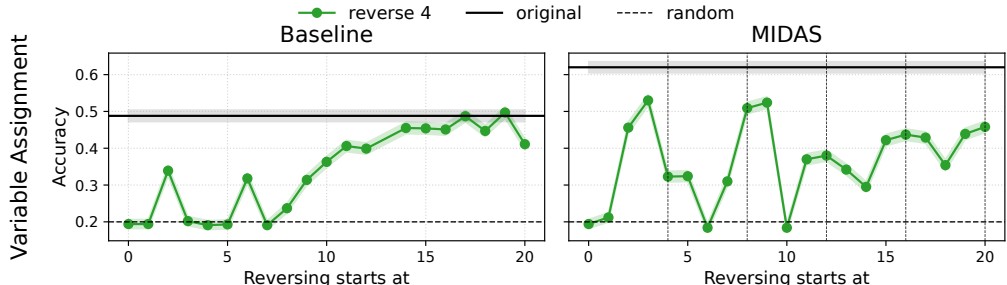

**Figure 6: Effect of reversing the order of four consecutive layers on reasoning primitive.** Reversing the order of layers within a block (first layer of each grown block as vertical grey lines; right figure) of size 4 degrades the performance for grown models more than swapping the same number of layers (len = 2 in Fig. 3). The baseline is more robust to reversing the order of the later layers, while MIDAS is especially sensitive to reversing the order across grown blocks, i.e., the last two and first two layers of consecutive blocks. Starting to reverse at these positions, which correspond to layer index 6, 10, and 14, always results in a substantial drop in performance. Appendix Fig. 11 shows results including LIDAS and an additional dataset.

**Experiments.** To measure the weight similarity of blocks for the grown model, we concatenate all weights from the feedforward layers of a block and calculate the cosine similarity to other blocks. Similarly to before, we skip layers and measure the relative change for all later layers, but now on *all* tokens (including the current token).

**Interpretation.** In LIDAS we observe a block-similarity structure that is symmetric about the model's centre, whereas in MIDAS the central block is more similar to the earlier (upper) blocks than to the later (lower) ones, yielding an asymmetric pattern (Fig. 7). This difference follows from the growth rule: LIDAS duplicates the exact layer-wise middle, while MIDAS is constrained to the nearest block centre. With an even number of blocks, the MIDAS choice necessarily biases similarity toward one side.

Additionally, this growing strategy leads to a higher utilisation of the first attention sublayer of every block (Fig. 7b), making it more aligned with the residual stream and having a bigger effect on the current computations of future layers. This effect is especially noticeable for deeper networks (Appendix Fig. 21), but we also observe it here for the 1.7B model with 24 layers.

## 5 DISCUSSION

**Mechanistic View.** The results suggest that gradual-depth growing qualitatively changes how Transformers use their depth. In contrast to standard training, where late layers can often be removed with a modest performance decrease due to the "Curse of Depth" (Sun et al., 2025b; Csordás et al.,

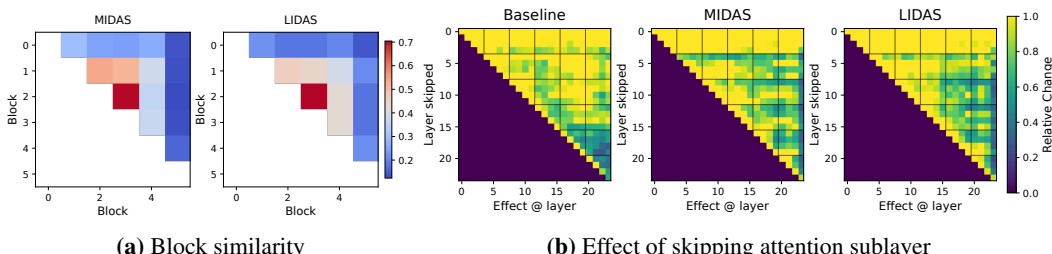

**(a)** Block similarity     **(b)** Effect of skipping attention sublayer

**Figure 7: Baseline vs. `MIDAS` vs. `LIDAS`.** (a) The weight similarity, measured by cosine similarity, between feedforward layers per block is more symmetric for `LIDAS` compared to `MIDAS`. We omit the baseline as its weight similarities are all close to zero. (b) Skipping the first attention sublayer of every block in the centre of the network has a lower effect on the following layers' *current* computations in `MIDAS` compared to `LIDAS`.

2025), depth-grown models allocate indispensable computation to later layers (Section 4.1). The layer-wise analyses indicate that layers in the middle of the network are not homogeneous, but organized into *permutable, block-wise computations* (Section 4.2) with an internal *cyclical (sub)layer structure* (Section 4.3). Taken together, the reported observations support a single picture: depth growth steers models toward learning a compact set of computational structures that are repeated along depth. This mirrors the spirit of Looped or Universal Transformers, which explicitly recursively apply a learned block of layers. However, here this loop-like behaviour emerges from the growth process without explicit weight tying. In this sense, the experiments support the hypothesis that depth grown models can be viewed as a relaxed version of Looped Transformers, where gradual growth (with layer duplication) steers the optimisation towards repeated computation without enforcing exact parameter sharing.

**`MIDAS` vs. `LIDAS`.** While `MIDAS` is motivated in Saunshi et al. (2024) by approximating the functional symmetries of Looped Transformers, its growing strategy is constrained by the the block size and yields an asymmetric weight similarity pattern. We propose `LIDAS` that modifies the growing strategy to duplicate at the layer-wise middle, restoring more symmetric weights similarities (Fig. 7a) and therefore being closer to the original motivation of `MIDAS`. Quantitatively, we observe that `LIDAS` often further amplifies the strengths of `MIDAS` by improving in reasoning-heavy tasks (MathWorld, Primitives), while negating its weaknesses by matching or exceeding the baseline in language modelling (NLL, Lambada, Hellaswag) at a 23% reduced training cost (Table 1). Qualitatively, `LIDAS` aligns its attention sublayers better to the residual stream compared to `MIDAS` (Fig. 4) making better use of the attention sublayers in the middle of the network (Fig. 7b).

# 6 CONCLUSION

This work systematically investigates how gradual depth growth in large language models affects their computational dynamics, providing a mechanistic explanation for their improved reasoning performance. Our findings confirm that gradually grown models, outperform conventionally trained baselines on reasoning tasks and in training cost. Through detailed analysis, we demonstrate that this performance is tied to a more effective utilization of model depth. Unlike non-grown models that suffer from a Curse of Depth (Sun et al., 2025b; Csordás et al., 2025), our grown models continue to perform novel computations in their later layers and exhibit a higher overall depth score. We show that this is enabled by the formation of permutable computational blocks in the middle of the network, with each layer within these blocks serving a distinct cyclical role. The superiority of our proposed lightweight and novel stacking variant `LIDAS` is attributed to its ability to create a more symmetric weight structure and more effective attention layers, leading to improved empirical results. In conclusion, our research provides critical insights into the internal workings of depth-grown models, confirming that these training procedures can overcome key architectural inefficiencies and pave the way for more efficient and capable model development.

## REPRODUCIBILITY STATEMENT

We conduct all training and experiments using publicly available code and datasets. Our training setup builds on the open-source `nanotron`[2] library; Table 2 lists all model and optimizer hyper-parameters, and Appendix A provides the exact SmolLM training mixture with links to each public dataset to fully reconstruct the training corpus. Additionally, we provide a detailed description of the growing operators in Section 3.1 and further detail in Appendix A. To reproduce our analyses, Appendix B details the evaluation protocols and the open-source libraries we use, along with any task-specific settings.

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

## DISCLOSURE OF LLM USAGE

Large language models were used for language editing, such as enhancing clarity, precision, and flow, and for minor aesthetic adjustments to figures to improve interpretability.

## A SMOLLM: ARCHITECTURE & DATA

**Data.** For all SmolLM models we trained, we followed the SmolLM-v1 data mixture from Ben Allal et al. (2024).

- **FineWeb-Edu (deduplicated)** (Ben Allal et al., 2024): Educational slice of FineWeb selected with a Llama3-70B–trained "educational quality" classifier. We use the deduplicated subset ($\approx$220B tokens) included in the SmolLM-Corpus.
- **OpenWebMath** (Paster et al., 2023): High-quality mathematical web pages extracted from Common Crawl with math-aware parsing, quality filtering, and deduplication ($\approx$14.7B tokens). Used to enrich math/reasoning coverage.
- **Cosmopedia v2** (Ben Allal et al., 2024): Synthetic textbooks, stories, and code generated with Mixtral-8×7B using curated topic lists and seed pages. v2 totals $\approx$39M documents ($\approx$28B tokens of textbooks/stories).
- **Python-Edu** (Ben Allal et al., 2024): Educational Python subset built by training an "educational code" classifier on annotated samples from *The Stack* and applying it to the Star-Coder training corpus. It contains $\approx$4B tokens with strict quality thresholding.

Given a fixed training-token budget, we then sample the corpus by proportion—70% FineWeb-Edu (deduplicated), 15% Cosmopedia v2, 9% Python-Edu and 6% OpenWebMath. Note that this leads to significant upsampling of the smaller datasets like Python-Edu and OpenWebMath.

**Model architecture.** Both sizes follow a LLaMA-style, decoder-only Transformer with RMSNorm, SwiGLU MLPs, and RoPE positional embeddings (tied input/output embeddings). The 360M variant uses GQA.

|  | SmolLM-1.7B | SmolLM-360M |
|---|---|---|
| Layers | 24 | 32 |
| Model width | 2048 | 960 |
| FFN dimension | 8192 | 2560 |
| Attention heads | 32 | 15 |
| KV heads | 32 (MHA) | 5 (GQA) |
| Norm | RMSNorm | RMSNorm |
| MLP activation | SwiGLU | SwiGLU |
| Batch size | 2M | 1M |
| Learning rate $\eta_{\max}$ | 0.0005 | 0.003 |
| Weight decay | 0.01 | 0.01 |
| Positional embeddings | RoPE ($\theta$=10,000) | RoPE ($\theta$=10,000) |
| Context length (pretrain) | 2048 | 2048 |
| Tokenizer | cosmo2 | cosmo2 |
| Tied embeddings | Yes | Yes |

Table 2: Hyperparameters for both SmolLM models

**Training.** We train both sizes for 200k iterations. This corresponds to roughly 200B seen tokens for the 360M model and 400B for the 1.7B model. We use a trapezoidal learning-rate schedule with a linear warmup for the first 2000 steps up to the peak rate $\eta_{\max}$, a constant plateau until step 170000, and a 1-sqrt decay over the final 30000 steps (Hägele et al., 2024). We optimise with AdamW (Loshchilov & Hutter, 2019) and apply global gradient clipping at 1.0 for all runs.

**Training with the Growing operator** For SmolLM with gradual depth growth all training hyperparameters match the baseline in Table 2. We use a fixed block size $b = 4$ and insert a new *middle* block after each stage, instantiating either MIDAS (duplicate the middle stage block) or LIDAS (duplicate the layer-wise middle; see Section 3.1), while keeping width and attention heads constant. At every growth step we deep-copy all layer parameters and their optimizer state so duplicated layers start identically (same AdamW moments) and then diverge with continued training; embeddings and the final head are copied unchanged. The number of growth stages is defined by $k = L_{\text{final}}/b$ and $T$ is the total number of training steps. We allocate per-stage budgets using the PROP-$\alpha$ schedule of Saunshi et al. (2024):

$$T_i \;=\; \frac{i^\alpha}{\sum_{j=1}^{k} j^\alpha}\, T \quad \text{for } i = 1, \ldots, k\,,$$

and use PROP-1 ($\alpha{=}1$) in our experiments. In practice, we round $T_i$ to integers (largest-remainder to keep $\sum_i T_i = T$) and maintain a *single continuous* learning-rate schedule across stages (no LR reset; the scheduler's global step carries over). We set $T = 170{,}000$ so all models reach their final depth before they enter the cooldown phase.

**Compute requirements.** We trained all models on NVIDIA A100 GPUs (40 GB). The large (static baseline) model ran on 128 GPUs for 4.5 days, and the small (static baseline) model ran on 64 GPUs for 1.5 days.

## B    EVALUATION SETUP

**Codebases.** Depth analyses and interventions follow the methodology of Csordás et al. (2025), which is extended to block-wise skip/swap operations over consecutive layers (block sizes $\{1, 2, 4, 8\}$) and further extended to permuting consecutive layers in arbitrary order. Tuned Lens experiments follow Belrose et al. (2023).

**Reproducibility defaults.** We adopt the default configuration from the original depth-analysis repository of Csordás et al. (2025) for reproducibility. Specifically, we use the same fixed set of GSM8K prompts/examples for early-exit, skip, swap and relative contribution evaluations, and we keep random seeds, batching, and evaluation hyperparameters at their defaults unless stated otherwise.

**Models and data.** We analyze SmolLM-v1 backbones at 360M and 1.7B parameters (training details in Appendix A) and evaluate on MATH, MQuAKE, and GSM8K as described in the main text. Preprocessing follows Csordás et al. (2025).

**Intervention protocols.** We distinguish *heatmap (relative-change) experiments* from *benchmarked interventions*. Heatmaps quantify relative changes and use **single (sub)layer skipping only**. Benchmarked interventions (accuracy-based) are described separately below. For heatmaps, we evaluate two intervention *modes* and two *measurement axes*, following and extending Csordás et al. (2025):

- **Current vs. future effects.** In the *current* setting, we intervene by erasing the entire (sub)layer contribution *for all tokens* and measure changes on all positions. In the *future* setting, for a chosen boundary token index $t$, we erase the (sub)layer contribution *only for tokens $\leq t$*, leaving tokens $> t$ unchanged at that (sub)layer; we then measure changes *strictly on tokens $> t$*. This design directly tests whether information is transferred to later tokens via attention, ruling out purely pointwise (self-only) computation.

- **Output vs. later-layer effects.** For the *output probability distribution*, we compute the L2 norm difference between the softmaxed logits of the intervened and original forward passes, aggregated over the relevant positions (current or future). For the *later-layer effects*, we compute, for each later layer, the *relative change* in the residual contribution (i.e., the norm of the difference in that layer's residual update divided by the norm of the original residual update), again aggregated over the relevant positions.

Concretely for heatmaps, in the *future* effects evaluation we select multiple boundary indices $t$ and, for each $t$, (i) erase the (sub)layer's contribution only at tokens $\leq t$, (ii) keep its contribution intact at tokens $> t$, and then compare the intervened and original runs on (a) softmaxed output distributions at positions $> t$ and (b) residual contributions of all later layers at positions $> t$. This directly tests whether features are moved forward in time (to future tokens) by attention.

For heatmaps, we also include a **local (direct) effects** variant, which isolates pairwise dependencies between a source layer and a later target layer without allowing effects to *propagate* through multiple subsequent layers. Specifically, for a source layer $s$ and a later layer $\ell > s$, we subtract the stored contribution of $s$ from the residual fed into $\ell$ and record the relative change at $\ell$; we do *not* roll this modification forward beyond $\ell$. This complements the propagated analyses by revealing direct, non-compounded influences.

Heatmap interventions are performed at the layer or sublayer level and are **strictly single-layer**. The current/future distinction applies *only* to these heatmap experiments. Block-wise operations are used solely in benchmarked interventions (below).

**Aggregation for heatmaps.** For heatmap visualizations of later-layer effects, we aggregate by taking the *maximum* relative change across (i) batch examples, (ii) eligible sequence positions, and (iii) multiple chosen boundaries $t$ in the future setting. Concretely, for current effects, we take the max over all positions; for future effects, we take the max only over positions strictly greater than $t$, and then take the max over all tested $t$ for each example. This yields a single matrix of source-layer by target-layer maxima per model/setting.

**Tuned Lens training and evaluation.** Following Belrose et al. (2023), we train, for each layer, a small affine adapter that maps that layer's residual output to the hidden representation with the same shape that serves as the input to the *final* normalization layer immediately before the unembedding. Final logits are then obtained by applying the model's final normalization and unembedding as usual. Adapters are trained on a held-out split of FineWeb-Edu and evaluated by (a) KL divergence between early-exit and final distributions and (b) top-5 vocabulary overlap with the final prediction (cf. Fig. 1B/C for 1.7B and Fig. 14B/C 360M).

**Benchmarked interventions.** We evaluate accuracy on downstream benchmarks under: (i) Tuned Lens early-exit (using the adapter path described above), (ii) skip interventions, and (iii) swap interventions. For benchmarks, we may intervene on contiguous **blocks** of sizes $\{2, 4, 8\}$ (in addition to single layers). We decode with greedy top-1 and compute benchmark accuracy (e.g., MathWorld, reasoning primitives), matching the evaluation protocol used for the unmodified model. The current/future distinction does *not* apply to benchmark evaluations.

**Depth score.** We report the *logit-effect* depth score based on `mean_dout`. For each layer $\ell$, `mean_dout` is the across-examples mean of the maximum L2 change in the softmaxed output distribution at future tokens when intervening at layer $\ell$ (future-setting; see intervention protocols). We normalize this per-layer vector to a probability distribution over layers and take its expected layer index as the depth score (Csordás et al., 2025).

## C    DETAILED BENCHMARK RESULTS

**Setup** In Section 3.2 we presented aggregated results over several Benchmarks. In this section, we show the detailed results for all models and for completeness, we also report results with `LN-Scaling` and further experiment with the combination of `LN-Scaling` and `LIDAS`/`MIDAS`. We evaluated all models on these benchmarks using the language model evaluation harness library (Gao et al., 2024).

**Reasoning Primitives** We implemented the **Reasoning Primitives** following the task descriptions in Saunshi et al. (2024). Induction copying is generated by sampling a sequence of random 3-letter words (e.g., length 10), selecting a contiguous subsequence (e.g., length 5) from within it, appending

that subsequence, and asking for the next token in the original sequence. Variable assignment is generated by sampling variable–value statements and querying a single variable's value. We use the same basic, math and code prompt templates.

An example for a *Copying random words* task would be:

*Prompt*:

```
    Fill in the blank:
    jic dqy sof uzg ewr oxw osp tkj rvw mnu jic dqy sof uzg ewr ___. ->
```

*Answer*:

```
    oxw
```

For an example of *variable assignment* task:

*Prompt*:

```
Fill in blank:

o=23
k=3
t=13
a=1
e=9
o=___. ->
```

*Answer:*

```
    23
```

Notice that for the above tasks multiple choice format is used and a 5-shot evaluation setting. This means that the random guessing baseline score is 10% for the Copying task and 20% for the variable assignment task.

| | | CoQA | DROP | QuAC | SquadV2 | TyDi QA (wc) | Mean |
|---|---|---|---|---|---|---|---|
| 360M | Baseline | 46.08 | 12.48 | 14.27 | 24.35 | 17.25 | 22.89 |
| | MIDAS | 50.00 | 12.75 | 14.10 | 24.96 | 21.06 | 24.57 |
| | LIDAS | 51.50 | **15.25** | **15.79** | 28.11 | **22.50** | **26.63** |
| | LN-Scaling | 44.80 | 13.18 | 12.45 | 24.11 | 21.14 | 23.14 |
| | LN-Scaling + MIDAS | 45.70 | 13.14 | 13.86 | 25.11 | 12.39 | 22.04 |
| | LN-Scaling + LIDAS | **53.17** | 13.03 | 14.59 | **30.69** | 16.19 | 25.5 |
| 1.7B | Baseline | 58.36 | 16.52 | 15.91 | 33.88 | 23.17 | 29.57 |
| | MIDAS | 59.35 | 16.88 | 17.30 | 36.06 | 14.39 | 28.80 |
| | LIDAS | **63.41** | 17.66 | **17.91** | **36.56** | 13.65 | **29.84** |
| | LN-Scaling | 54.94 | **17.84** | 16.61 | 32.78 | **23.37** | 29.11 |
| | LN-Scaling + LIDAS | 62.36 | 16.09 | 17.74 | 34.98 | 9.05 | 28.04 |

**Table 3: Open-book QA Benchmarks**

.

**Results** In Tables 3 and 4 we report per-dataset results for Open-Book and Closed-Book QA. In line with Saunshi et al. (2024), both grown models (MIDAS and LIDAS) yield larger gains on Open-Book QA than on Closed-Book QA. Notably, LIDAS 1.7B improves over the 1.7B baseline even on most Closed-Book datasets and remains competitive on the rest, which differs from observations made for MIDAS in Saunshi et al. (2024). Overall, the grown variants confer modest but consistent Open-book gains, whereas LN-Scaling alone yields only small changes relative to the non-grown baseline. Combining growing with LN-Scaling can sometimes improve over the standard LN-Scaling setting, but often still falls short when compared to LIDAS.

| | | Trivia QA | Web Questions | TyDi QA (nc) | Natural Questions | Mean |
|---|---|---|---|---|---|---|
| 360M | Baseline | 19.23 | **16.78** | 12.98 | 9.01 | 14.50 |
| | MIDAS | 18.90 | 14.58 | 12.64 | 8.89 | 13.75 |
| | LIDAS | **20.80** | 15.61 | 12.14 | 9.73 | 14.57 |
| | LN-Scaling | 20.52 | 16.73 | 12.36 | **9.94** | **14.89** |
| | LN-Scaling + MIDAS | 17.76 | 16.60 | 12.66 | 9.18 | 14.05 |
| | LN-Scaling + LIDAS | 18.23 | 15.32 | **13.67** | 9.26 | 14.12 |
| 1.7B | Baseline | 27.72 | 19.20 | 15.34 | 12.18 | 18.61 |
| | MIDAS | **27.98** | 17.96 | 16.16 | 11.91 | 18.50 |
| | LIDAS | 26.85 | 20.24 | **16.34** | **12.90** | **19.08** |
| | LN-Scaling | 25.12 | **20.89** | 15.71 | 12.79 | 18.63 |
| | LN-Scaling + LIDAS | 25.88 | 20.15 | 15.78 | 11.88 | 18.42 |

**Table 4: Closed-book QA Benchmarks**.

| | | ASDiv | MAWPS Add/Sub | MAWPS Multi-Arith | MAWPS Single-Op | MAWPS Single-Eq | SVAMP | Mean |
|---|---|---|---|---|---|---|---|---|
| 360M | Baseline | 3.34 | 3.67 | **1.72** | 5.66 | 2.75 | 5.02 | 3.69 |
| | MIDAS | 3.77 | 3.67 | 1.15 | 6.29 | 6.42 | 5.02 | **4.39** |
| | LIDAS | **4.64** | 1.83 | **1.72** | **7.55** | 6.42 | 4.01 | 4.36 |
| | LN-Scaling | 2.95 | 0.00 | 1.15 | 4.40 | 5.50 | 3.34 | 2.89 |
| | LN-Scaling + MIDAS | 3.56 | **5.50** | 1.15 | 1.89 | 6.42 | **5.69** | 4.04 |
| | LN-Scaling + LIDAS | 3.21 | 3.67 | **1.72** | 4.40 | **7.34** | 3.68 | 4.00 |
| 1.7B | Baseline | 11.15 | 14.68 | 1.15 | 25.16 | **22.02** | 8.36 | 13.75 |
| | MIDAS | 12.93 | 18.35 | 2.30 | 33.96 | 20.18 | 8.70 | 16.07 |
| | LIDAS | **14.88** | **25.69** | 2.87 | **38.36** | 18.35 | 11.37 | **18.59** |
| | LN-Scaling | 10.20 | 13.76 | 1.72 | 17.61 | 14.68 | 8.03 | 11.00 |
| | LN-Scaling + LIDAS | 14.19 | 22.02 | **3.45** | 31.45 | 21.10 | **11.71** | 17.32 |

**Table 5: Math World**.

On the reasoning benchmarks, Math World (Table 5) and Reasoning Primitives (Table 6), improvements at 360M are modest on average, while at 1.7B they become more pronounced. For Math World, LIDAS 1.7B attains the best score on five out of six benchmarks (the exception is MAWPS Single-Equation). For Reasoning Primitives, both MIDAS and LIDAS surpass the baseline, with LIDAS 1.7B leading on copying and on the code/math variable-assignment formats, while MIDAS slightly edges LIDAS on the basic variable-assignment format. We notice however, that the variance of performance between tasks is much higher compared to language tasks.

| | | Copying (random words) | Copying (real words) | Variable assignment (basic) | Variable assignment (code) | Variable assignment (math) | Mean |
|---|---|---|---|---|---|---|---|
| 360M | Baseline | 15.50 | 13.30 | 20.50 | 58.30 | 42.70 | 30.06 |
| | MIDAS | 13.80 | 14.10 | 20.00 | 52.90 | 40.10 | 28.18 |
| | LIDAS | 14.20 | 19.70 | 24.30 | 51.80 | 46.00 | 31.20 |
| | LN-Scaling | 17.90 | 16.40 | 22.70 | 49.10 | 50.80 | 31.38 |
| | LN-Scaling + MIDAS | 15.80 | 16.00 | **26.30** | **58.70** | 48.40 | 33.04 |
| | LN-Scaling + LIDAS | **21.90** | 19.80 | 25.50 | 57.20 | **52.10** | **35.30** |
| 1.7B | Baseline | 16.80 | 23.60 | 20.80 | 64.20 | 48.80 | 34.84 |
| | MIDAS | 19.30 | 24.60 | **37.00** | 61.50 | 62.00 | 40.88 |
| | LIDAS | **28.40** | **31.00** | 36.70 | 71.80 | 68.80 | **47.34** |
| | LN-Scaling | 24.40 | 26.60 | 23.80 | **75.20** | **71.90** | 44.38 |
| | LN-Scaling + LIDAS | 17.00 | 23.00 | 34.80 | 71.20 | 70.60 | 43.32 |

**Table 6: Reasoning Primitives**.

Consistent with our depth analyses, these benchmark trends coincide with higher depth scores and later-layer reliance for grown models, whereas LN-Scaling in our setup does not increase depth utilization relative to the baseline nor improve performance.

| Model | PetaFLOPs | Ratio |
|---|---|---|
| 360M Standard | 613527.488 | 1.289 |
| 360M Grown | 476147.897 | 1.000 |
| 1700M Standard | 4813222.102 | 1.288 |
| 1700M Grown | 3736608.182 | 1.000 |

**Table 7: PetaFLOPs used for training 200k iterations with block size of 4 and PROP-1 growing schedule.**

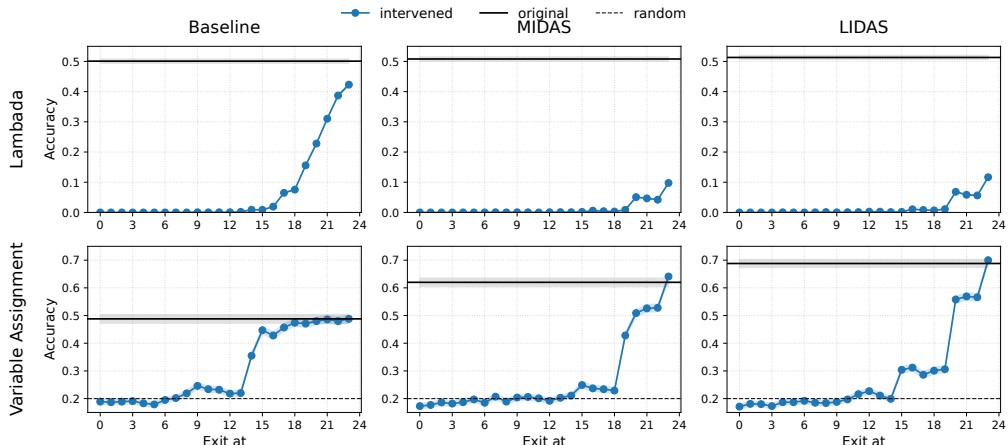

**Figure 8:** Early exit with tuned lens on *Lambada* and *Variable Assignment Math* for the Baseline, `MIDAS`, and `LIDAS` models at scale 1.7B.

In addition to improved reasoning performance, models trained with gradual stacking also require fewer computational resources, as reported in previous works. Specifically, `MIDAS` and `LIDAS` only need $\approx 77\%$ of the FLOPs used to train the baseline in Table 7.

# D ADDITIONAL RESULTS ON DEPTH ANALYSIS

In Section 4 we have shown how growing can alter the structure within the model, leading to better depth utilisation and different altering of the residual stream and robustness. However, these results have mainly been presented for `MIDAS` on the 1.7B model scale. In this section we want to first add results for `LIDAS`, to show that it does exhibit the same patterns as `MIDAS` and also show results on the smaller 360M scale. Finally, we include additional ablation plots for block sizes different from 4.

## D.1 1.7B MODELS

This section extends the main analyses for the 1.7B models to `LIDAS`. For each hypothesis made in Section 4, we show that our results also hold for `LIDAS` by resuming each experiment.

**Additional results for Section 4.1** Fig. 8 reproduces the early-exit analysis at 1.7B on Lambada and Variable Assignment, showing that grown models' early exit performance is relatively poor over the entire stack while the baseline saturates much earlier. Notably, this result is stable across tasks with different absolute accuracies, suggesting that reliance on later layers reflects a training-induced computational pattern rather than task difficulty. This complements the Section 4.1 diagnostics and reinforces that depth growth yields genuinely deeper computation at scale.

**Additional results for Section 4.2** Fig. 9 extends the swap-ablation result (Fig. 3) to include `LIDAS` at 1.7B. Grown models are markedly more robust than the baseline when swapping multi-layer blocks (sizes 2–8), especially in the middle of the network, conforming to the signature of

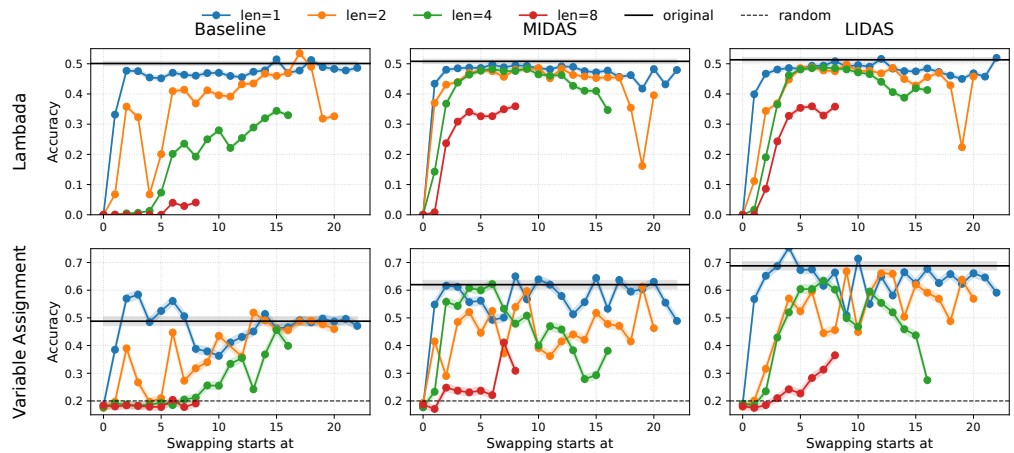

**Figure 9:** Swap ablations on *Lambada* and *Variable Assignment Math* for the Baseline, `MIDAS`, and `LIDAS` models at scale 1.7B.

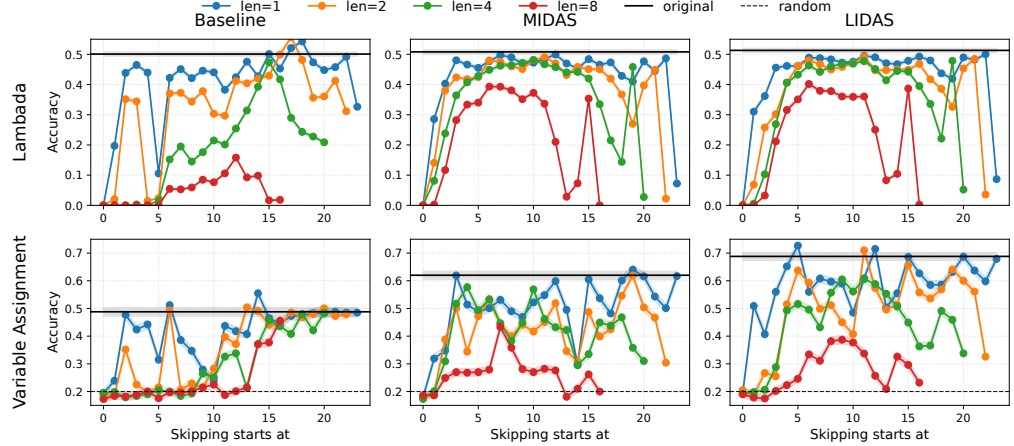

**Figure 10:** Skip ablations on *Lambada* and *Variable Assignment Math* for the Baseline, `MIDAS`, and `LIDAS` models at scale 1.7B.

block-level permutability argued in Section 4.2. Fig. 10 provides the complementary experiment, in which we skip consecutive layers of different sizes. Together, these interventions support our hypothesis that depth growth organizes computation into mid-network blocks whose presence is crucial but whose internal order is comparatively flexible.

**Additional results for Section 4.3** Figure Fig. 11 shows that grown models are particularly fragile when reversing four-layer windows that pass block boundaries, degrading more than under swapping or skipping, indicating that while blocks are permutable as units, the intra-block progression encodes roles that do not commute, as argued in Section 4.3. We can clearly see that `LIDAS` also follows this pattern and that it is independent of the task being considered. Figs. 12 and 13 reveal repeating mid-block motifs and stronger downstream propagation from later layers in grown models, generalising the results from Fig. 5 to include `LIDAS` at 1.7B. Collectively, these results support that our claims from Section 4.3 do hold for `LIDAS` and are not task-specific.

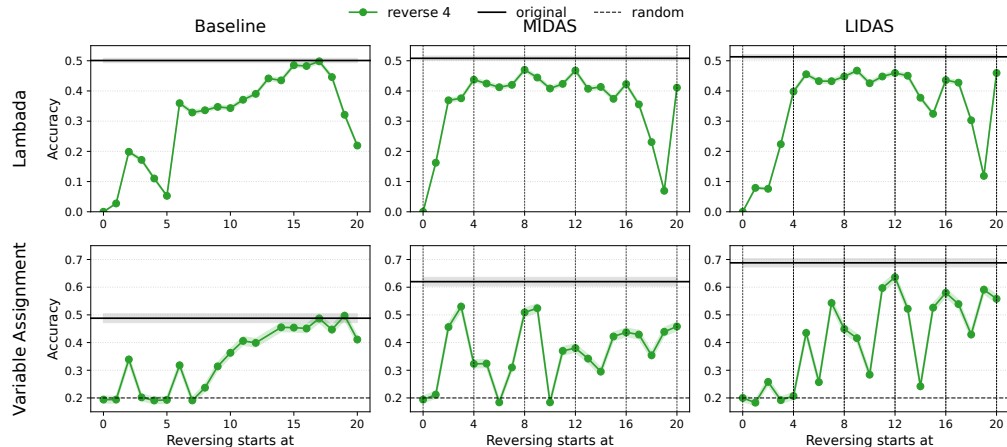

**Figure 11:** Reversing the order of 4 consecutive layers on *Lambada* and *Variable Assignment Math* for the Baseline, `MIDAS`, and `LIDAS` models at scale 1.7B.

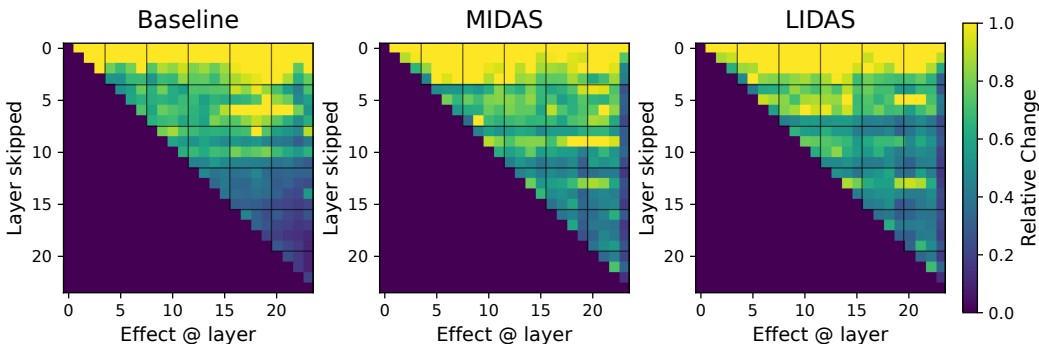

**Figure 12:** Propagated future effects of single-layer skipping for the Baseline, `MIDAS`, and `LIDAS` models at scale 1.7B.

### D.2   360M MODELS

This section presents the depth analysis results for the 360M models for the experiments described in the main paper.

**Additional results for Section 4.1**   In Fig. 14, we summarize 360M model depth utilization using the depth score and tuned-lens early-exit diagnostics (see Fig. 1 for the 1.7B case). In Fig. 14 A, we observe that the results are more task dependent compared to the 1.7B model: for experiments conducted on the MATH dataset, the trend of grown models to utilize more depth is evident (although to a lesser extent compared to the 1.7B model). However, for the MQuAKE dataset, the depth utilization pattern appears more complex, as only `MIDAS` has a higher depth score than the baseline, and no clear conclusion can be derived. In Fig. 14 B, early exiting for baseline model saturates much earlier, consistent with the 1.7B case. Moreover, in Fig. 14 C we can also see that pick performance is reached earlier for baseline compared to `MIDAS` and `LIDAS`.

In Fig. 15 we replicate benchmarked early exit interventions at 360M: again the picture is less clear compared to the 1.7B model case and the results are task dependent; in Lambada early exiting performance for grown models stays close to 0 until the very end, indicating the necessity of later layers in information processing. In Variable Assignment, however, `MIDAS` and `LIDAS` exhibit different behaviour when compared to the baseline.

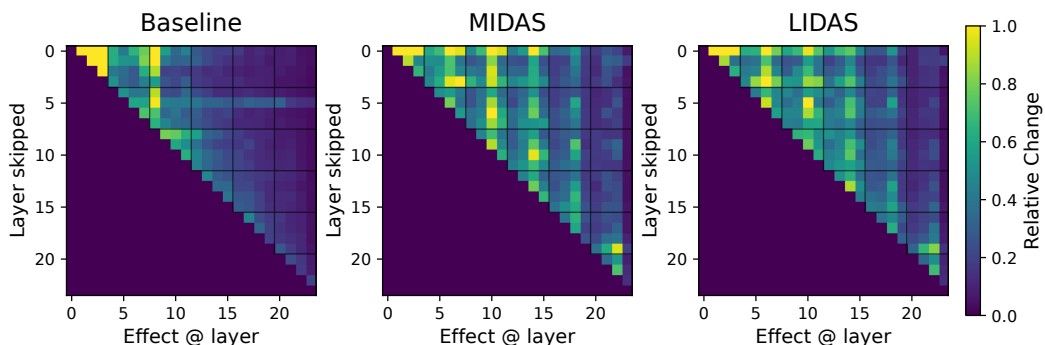

**Figure 13:** Local future effects of single-layer skipping for the Baseline, MIDAS, and LIDAS models at scale 1.7B.

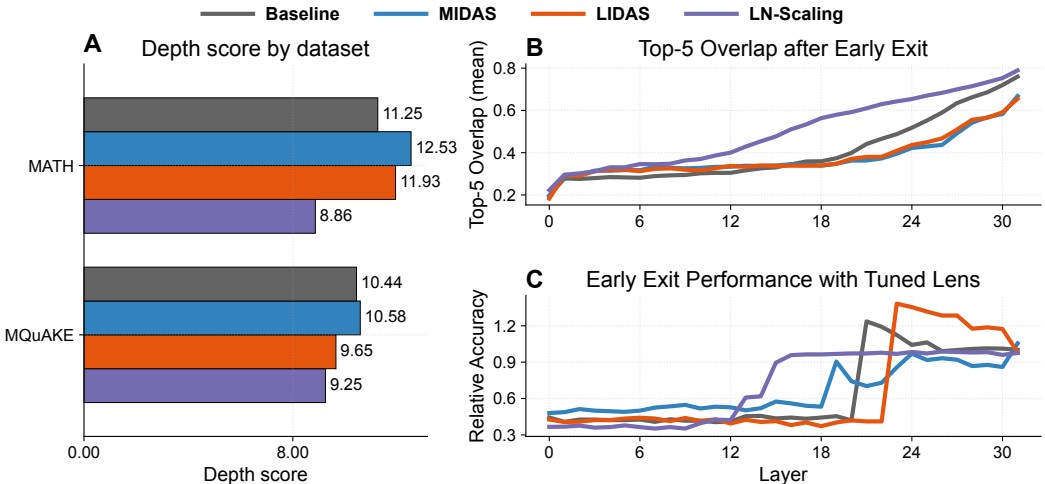

**Figure 14: Depth-grown models use their depth more (360M).** (A) Depth score (Csordás et al., 2025) on MATH (Hendrycks et al., 2021) and MQuAKE (Zhong et al., 2023). Grown models (MIDAS, LIDAS) have consistently higher depth scores, except LIDAS on MQuAKE. (B) Top-5 overlap between each layer's early-exit vocabulary and model's final vocabulary on 20 prompts from GSM8K (Cobbe et al., 2021). Both grown models studied in this work exhibit lower overlap at later layers, indicating that these later layers still contribute additional features necessary for the final prediction. (C) Early-exit relative accuracy versus layer on *Variable Assignment Math* reasoning primitive. The baseline reaches its best performance early, whereas accuracy for MIDAS and LIDAS is the highest at later layers. Using these metrics, however, LN-Scaling shows no discernible benefit over the baseline in depth utilisation.

**Additional results for Section 4.2** In Figs. 16 to 18 we assess robustness under reduced capacity, covering swap, skip, and reversal interventions. Consistent with the 1.7B case, the swap experiments show higher robustness w.r.t block level ordering interventions for the grown models. Moreover, when it comes to the reverse ordering interventions, we observe again this increased sensitivity of the grown models wrt block boundaries.

**Additional results for Section 4.3** We include the small-model counterparts of the future propagated, future local, and current attention ablations (see Figs. 19 to 21). In a nutshell, all cyclical patterns observed for the 1.7B model in the main paper still hold for the 360M case: the sensitivity of the 3rd layer of each block to the output of all its previous layers Figs. 19 and 20 and also reduced impact of the first attention sublayer of every block to later layers for MIDAS model. Notice that these effects are even more pronounced for the 360M model compared to the 1.7B case (compare the corresponding light and dark stripes in Figs. 20 and 21 to these in Figs. 7b and 13)

For completeness, we also show block-similarity structure at 360M (Fig. 22), which mirrors the symmetry patterns observed at 1.7B (cf. Fig. 7a).

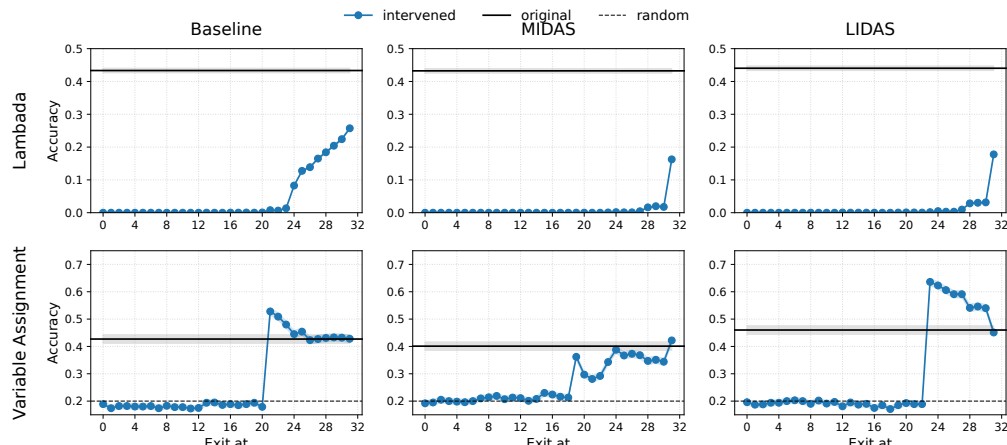

**Figure 15:** Small models (360M): early exit with tuned lens on *Lambada* and *Variable Assignment Math* for Baseline, MIDAS, and LIDAS.

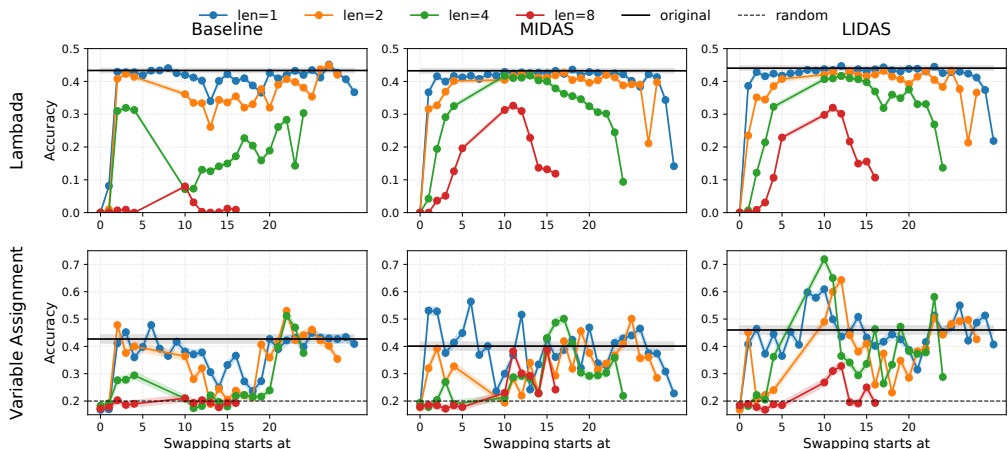

**Figure 16:** Small models (360M): swap ablations on *Lambada* and *Variable Assignment Math*.

### D.3 ABLATING BLOCK SIZE

We report mean relative contribution and cosine similarity plots for two ablated model settings: 360M models with block size 8 (Fig. 23) and 1.7B models with block size 3 (Fig. 24). As in Fig. 4, we observe clear patterns throughout block computation: for the 1.7B MIDAS and LIDAS models, both mean relative contribution and cosine similarity peak at the last layer of every block, and in the 360M grown models cosine similarity is likewise maximized at the final layer of each block. These experiments suggest that the emergence of such patterns is a characteristic of the growing training method itself, rather than a peculiarity of a specific block size.

## E LAYER NORM SCALING

LayerNorm-Scaling (LN-Scaling Sun et al. (2025b)) is a method that modifies the layer norm sublayer of pre-layernorm transformer architectures with the purpose of increasing the depth usage of later layers. It scales (the variance of) the output $h'_l$ of the layer normalization inversely by the square root of its depth

$$h'_l = \frac{1}{\sqrt{l}}\text{LayerNorm}(h_l)$$

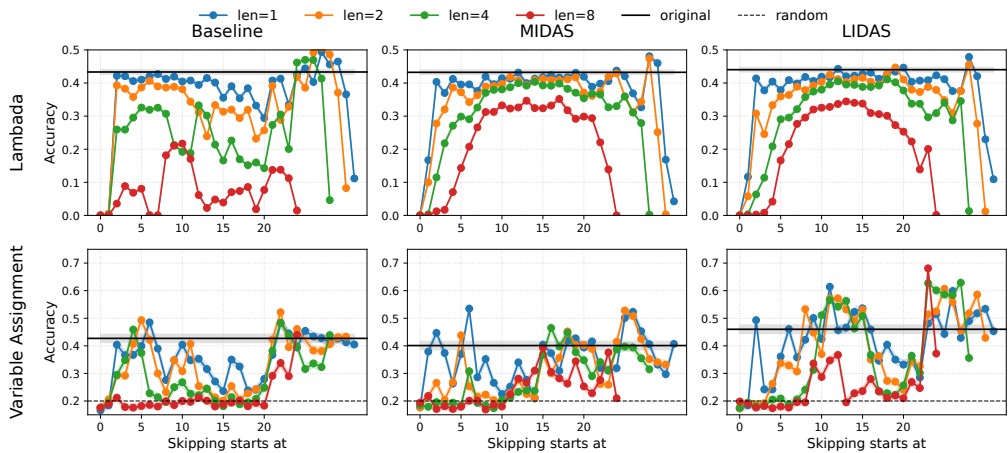

**Figure 17:** Small models (360M): skip ablations on *Lambada* and *Variable Assignment Math*.

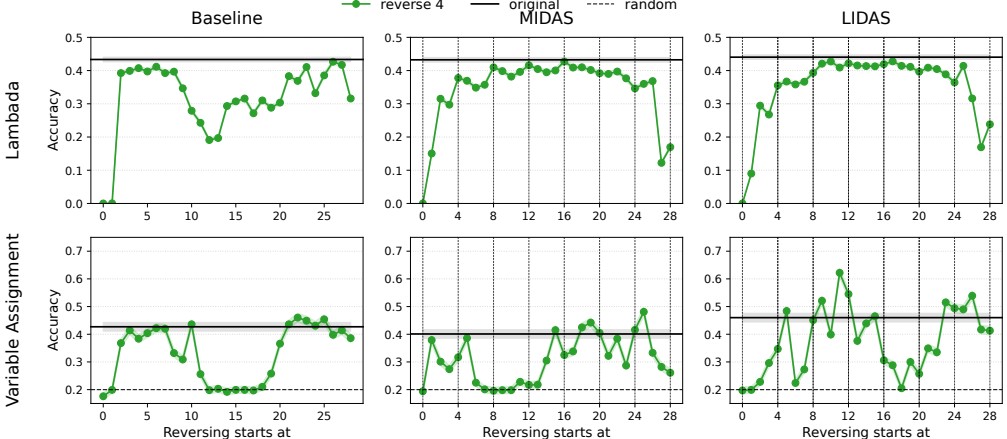

**Figure 18:** Small models (360M): reversing the order of 4 consecutive layers on *Lambada* and *Variable Assignment Math*.

where $h_l$ is the input of the layer norm sublayer. This simple modification mitigates the output variance explosion of deeper Transformer layers, improving their contribution. Additionally, it preserves the training stability common to all pre-layernorm models, which is demonstrated both theoretically and experimentally.

In our setting, however, LN-Scaling does not improve depth utilization according to the three diagnostics we used in Fig. 1. At both scales, the depth score shifts earlier, top-5 early-exit overlap increases at earlier layers, and tuned-lens early-exit accuracy plateaus sooner than for the baseline and grown models (Figs. 1 and 14).

To probe this further, we evaluate *future-propagated*, *future-local*, and *current-attention* ablations under LN-Scaling (see Appendix B for definitions of these interventions). Across both 1.7B (Fig. 25) and 360M (Fig. 26) models, interventions to later layers produce smaller downstream effects than in the Baseline, LIDAS, and MIDAS models, supporting our earlier finding that LN-Scaling concentrates computation earlier rather than improving later-layer usage. In line with Table 1, the apparent effectiveness of LN-Scaling diminishes at larger scales. This scale sensitivity may explain the discrepancy with Sun et al. (2025b), which does not scale to larger settings.

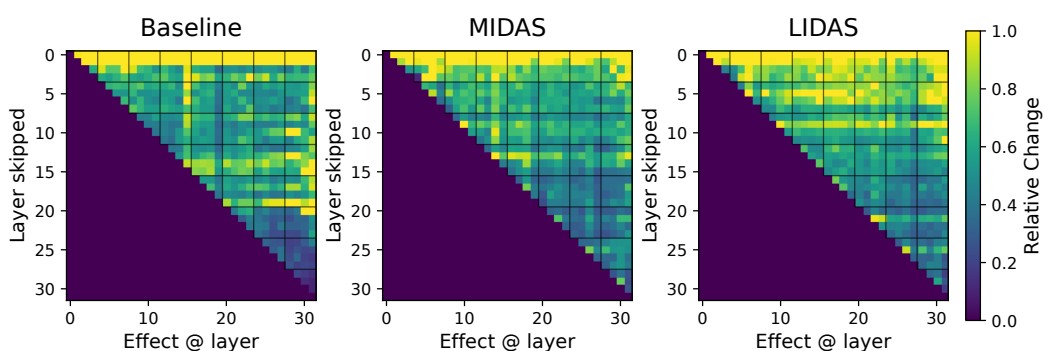

**Figure 19:** Small models (360M): propagated future effects of single-layer skipping.

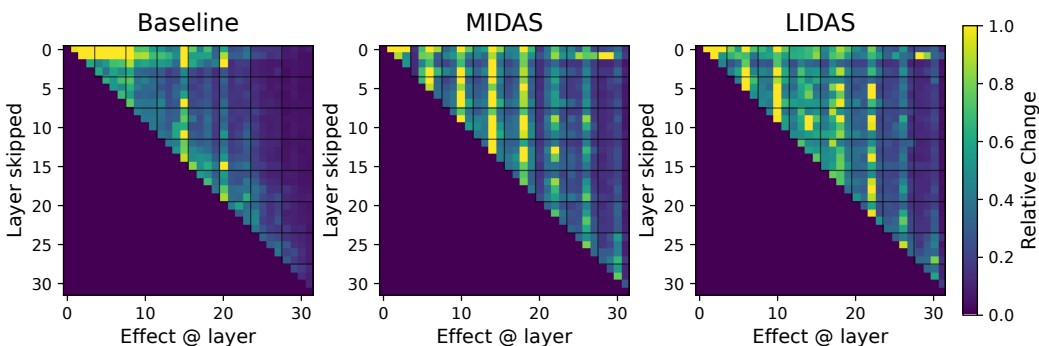

**Figure 20:** Small models (360M): local future effects of single-layer skipping.

To understand if `MIDAS` and `LIDAS` are also effective for this new architecture, we investigated the depth utilization when combining `LN-Scaling` and growing. In Figs. 27 and 28 we find, that growing can also increase the depth usage of architectures using `LN-Scaling`, indicating the generality of our findings.

Furthermore, we find that LayerNorm-Scaling does not yield substantial gains over the baseline and is typically outperformed by LIDAS, particularly at the 1.7B model size. When combining LayerNorm-Scaling with LIDAS Table 8, we observe consistent improvements over the LayerNorm-scaled baseline for the 360M model in all categories except Closed-book Q&A, and in reasoning-heavy tasks also for the 1.7B model.

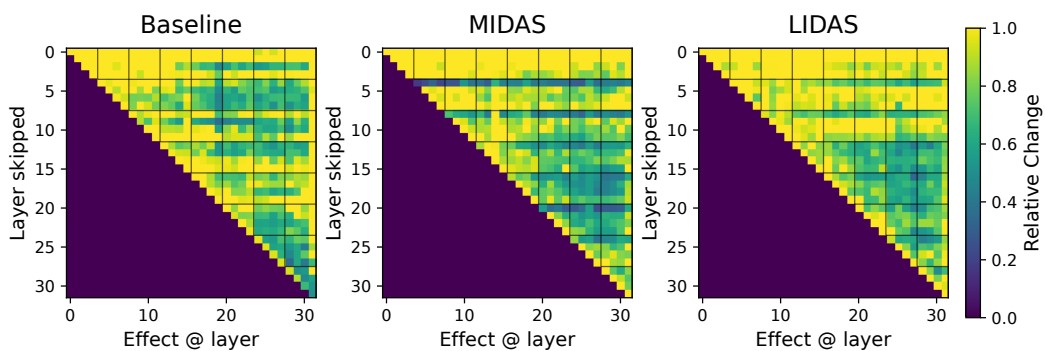

**Figure 21:** Small models (360M): current effects when skipping the attention sublayer.

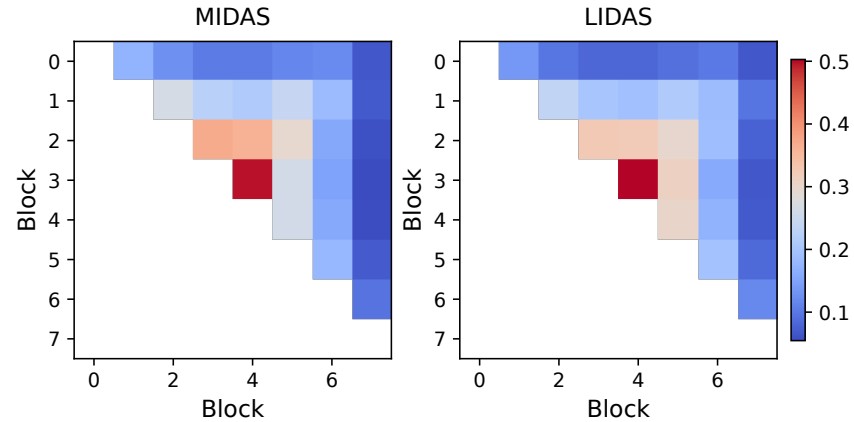

**Figure 22:** Small models (360M): block similarity for `MIDAS` and `LIDAS`.

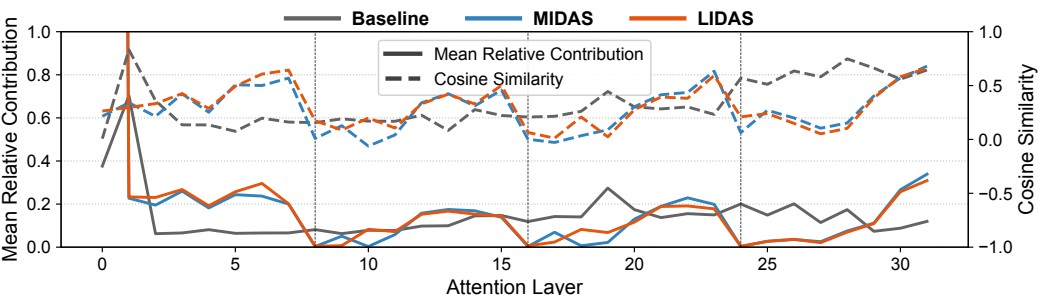

**Figure 23:** 360M models with block size 8: Mean Relative Contribution and Cosine Similarity plots.

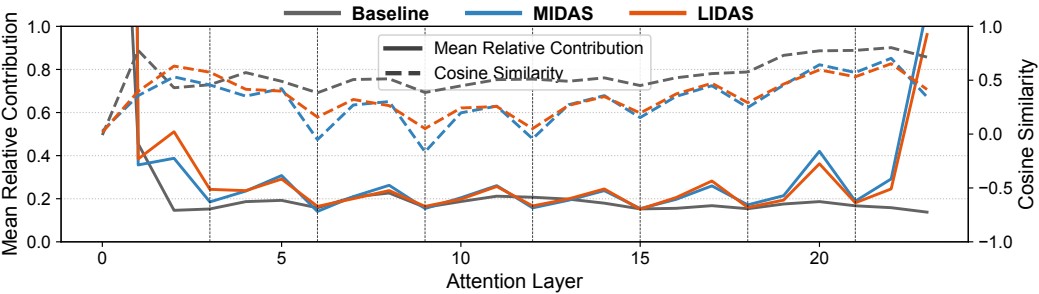

**Figure 24:** 1.7B models with block size 3: Mean Relative Contribution and Cosine Similarity plots.

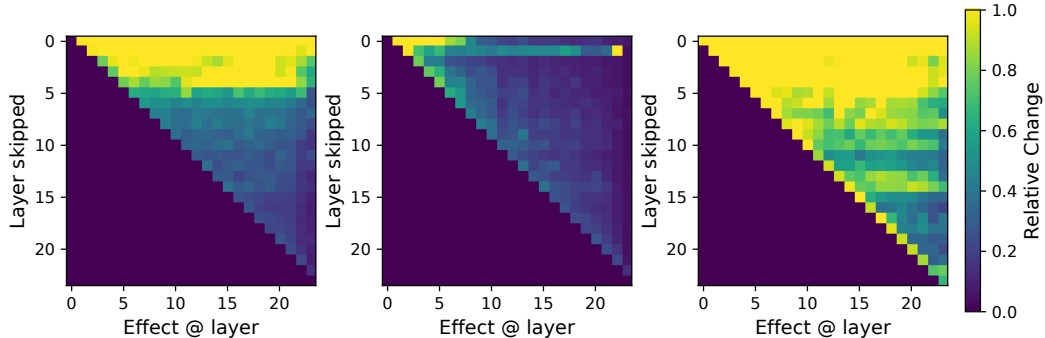

**Figure 25:** Baseline (1.7 B) model combined with `LN-Scaling`: (from left to right) future propagated layer, future local layer and current attention effects heatmaps

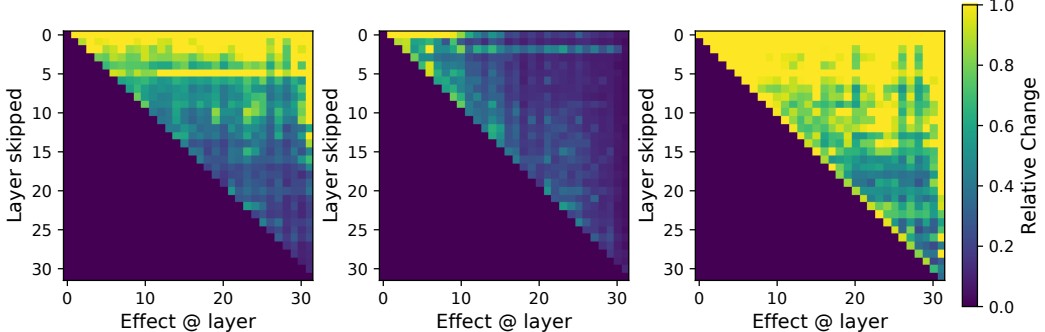

**Figure 26:** Baseline (360 M) model combined with `LN-Scaling`: (from left to right) future propagated layer, future local layer and current attention effects heatmaps

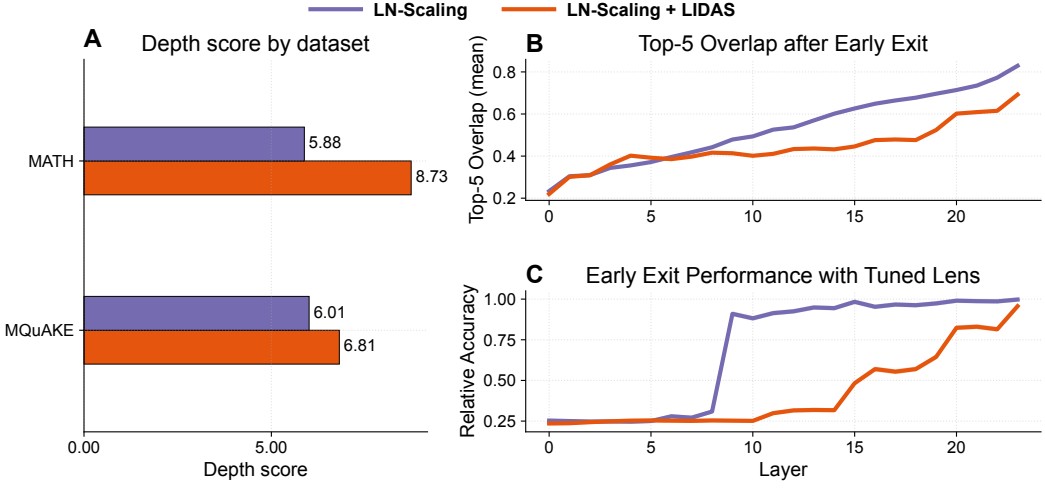

**Figure 27: Depth-grown models use their depth more also when applying growing techniques to `LN-Scaling` (1.7B).**

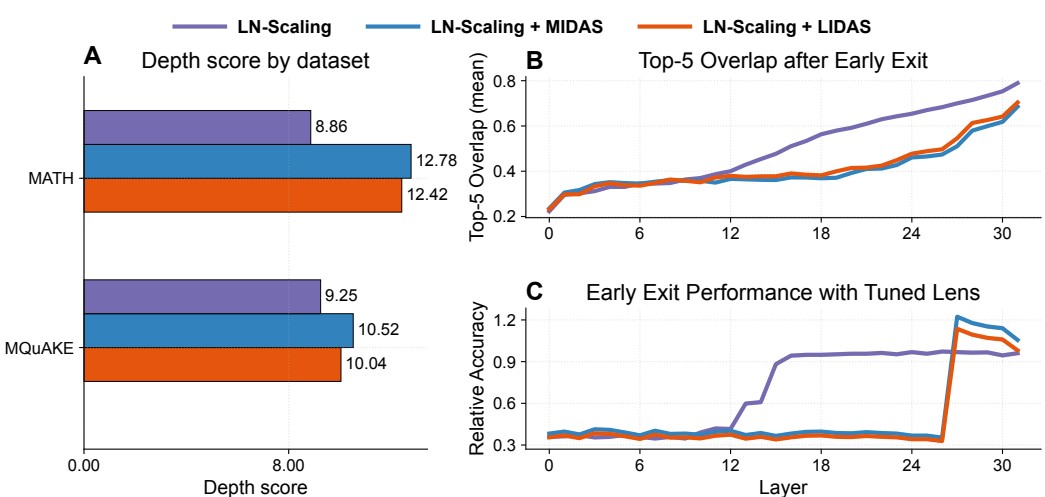

**Figure 28: Depth-grown models use their depth more also when applying growing techniques to `LN-Scaling` (360M)**

| | | | Standard cooldown | | | | | | Math cooldown | |
|---|---|---|---|---|---|---|---|---|---|---|
| | | **Holdout Set** (NLL ↓) | **Open-book Q&A** (F1 ↑) | **Closed-book Q&A** (F1 ↑) | **Lambada** (Acc ↑) | **Hellaswag** (Acc ↑) | **MathWorld** (Acc ↑) | **Primitives** (Acc ↑) | **MathWorld** (Acc ↑) | **Primitives** (Acc ↑) |
| 360M | LN-Scaling | **2.16** | 23.13 | **14.89** | 42.17 | 40.00 | 2.89 | 31.38 | 8.45 | 41.26 |
| | LN-Scaling + MIDAS | 2.19 | 22.04 | 14.05 | 42.77 | 39.84 | **4.03** | 33.04 | 7.72 | 36.90 |
| | LN-Scaling + LIDAS | **2.16** | **25.54** | 14.12 | **44.83** | **41.06** | 4.00 | **35.30** | **12.43** | **53.48** |
| 1.7B | LN-Scaling | 1.97 | **29.11** | **18.63** | 48.94 | 45.45 | 11.00 | **44.38** | 17.84 | 50.58 |
| | LN-Scaling + LIDAS | **1.96** | 28.04 | 18.42 | **51.37** | **46.69** | **17.32** | 43.32 | **23.98** | **56.28** |

**Table 8: Downstream performance of baseline and depth–grown models under `LN-Scaling`.** Compared to Table 1, `LIDAS` typically improves over the baseline, especially on reasoning-heavy tasks, but the gains are not uniform across datasets, indicating that growth combined with `LN-Scaling` yields architectures with qualitatively distinct behaviour.