# OpenReview forum: "Do Depth-Grown Models Overcome The Curse Of Depth? An In-Depth Analysis"
_ICLR.cc/2026/Conference — Submitted to ICLR 2026_

### Official Review · Reviewer_UGkv · 2025-10-15

**Soundness:** 3
**Presentation:** 4
**Contribution:** 2
**Rating:** 4
**Confidence:** 4

**Summary:**

This paper performs an analysis of MIDAS a gradual stacking technique to grow transformers whilst training them. The paper reproduces the MIDAS experiments on SmolLM at 360M and 1.7B scales.
The paper introduces LIDAS, slightly different to MIDAS, as when intruding a new 4 layer block B in-between B’ and B’’, takes the last two layers of B’ and first two layers of B’’. Where as MIDAS would just set B=B’. I view LIDAS less as a flashy novel method and more of a sanity check for the authors analysis experiments, although the authors find LIDAS is competitive with MIDAS.

The paper performs an in depth analysis of MIDAS and LIDAS:
1. Depth grown transformers utilise their depth more efficiently
2. Depth grown models are more robust to block level reordering but less robust to layer level reordering or removal.
3. Block grown models exhibit cyclic patterns within the layers
4. MIDAS and LIDAS weights are more symmetric.

**Strengths:**

- The paper independently reproduces prior results, something often overlooked in ML research
- The papers presentation is very clear.
- The analysis presented is deep and through.

**Weaknesses:**

- The analysis is limited to one model, SmolLM-1.7b. Although this is an expensive analysis to perform.
- The benchmarks used are a little limited, perhaps the FineWeb-Edu benchmarks could be useful here.
- Initialisation and learning rate scheme not described for baseline, this is known to impact a models ability to use deeper layers effectively (https://arxiv.org/pdf/2505.01618).

Minor: The figures are slightly out of order with the text, e.g. Figure 1 is a long way away from where it is described. Reorganising slightly would make the presentation perfect.

**Questions:**

1. What learning rate and initialisation scaling methods were used during training? For example, Dey et al. (https://arxiv.org/pdf/2505.01618) show better learning rate scaling can lead to more effective training.
2. Figure 1 shows that depth grown models use their later layers, but this needed? For example it now looks like earlier layers are used less, does growing move the problem or solve the problem?
3. The paper discusses weights being more symmetric when using M/LIDAS, do we necessarily want symmetric weights in our trained language models?
4. Is there a link between M/LIDAS vs regular training and how pruning of weights post training can be conducted? For example, regular training may allow practitioners to prune more efficiently?

In the rebuttal I would be most interested in hearing about the initialisation and learning rate scaling for the baseline, then answers to the above questions.

---

> ### Author Response · Authors · 2025-11-18
>
> We thank the reviewer for their insightful remarks and for recognising the clarity (“The paper's presentation is very clear”) and soundness (“The analysis presented is deep and thorough”) of the presentation. We address the specific points raised below. Please see the general response above, which provides a concise summary of the principal updates made in the revised manuscript.
>
>
> ## Weaknesses
>
> **The analysis is limited to one model, SmolLM-1.7b. Although this is an expensive analysis to perform.**
>
> We thank the reviewer for acknowledging the cost of this analysis and agree that demonstrating robustness across architectures helps establish the generality of mechanistic claims. While Section 4 focuses on SmolLM-1.7B, this analysis is not restricted to a single model size. As detailed in Appendix D, we repeat all depth analyses for a 360M SmolLM variant and observe the same qualitative phenomena. However, the ability to perform analyses beyond one model family is limited due to numerous reasons: (1) In order to achieve a fair comparison, we pretrain all models from scratch on fixed token budgets. Therefore, we chose to focus on a widely adopted Pre‑LN setup in SmolLM, which follows “Llama” model configurations and enables controlled analysis under a constrained budget. (2) We focus on the scaling axis, showing the generality of the claims across model scales. Running these expensive experiments on multiple architectures & scales is, unfortunately, beyond the computational possibilities of any academic and most industrial labs.
>
> To address this concern, however, we present additional results on models utilising LayerNorm-Scaling (LN-Scaling; [1]) as an architectural variant in Appendix E. We have trained both MIDAS and LIDAS with LN-Scaling and again find the same trends in depth utilisation and reasoning performance. This further supports the claim that the findings are not an artefact of a single model instance but generalise across model size and variant architectures.
>
> **The benchmarks used are a little limited, perhaps the FineWeb-Edu benchmarks could be useful here.**
>
> The main text reports aggregated results over 21 benchmarks (Table 1). The full per-dataset breakdown (including NLL) is provided in Tables 3–6 in Appendix C. Nevertheless, we agree that adding further evaluation would be valuable and provide additional results on the FineWeb-Edu benchmarks using the 1.7B models.
>
> | Task | Baseline | MIDAS | LIDAS | LN-Scaling |
> |---|---|---|---|---|
> | arc challenge | 0.409 | **0.414** | 0.412 | 0.413 |
> | arc easy | **0.715** | 0.683 | 0.708 | 0.701 |
> | commonsenseqa | 0.353 | 0.356 | **0.363** | 0.306 |
> | hellaswag | 0.593 | 0.598 | **0.599** | 0.574 |
> | openbookqa | **0.418** | 0.416 | 0.412 | 0.404 |
> | piqa | 0.754 | 0.745 | **0.760** | 0.752 |
> | siqa | 0.413 | 0.405 | 0.406 | **0.424** |
> | winogrande | 0.542 | **0.562** | 0.559 | 0.561 |
> | mmlu average | **0.378** | 0.374 | 0.371 | 0.367 |
> | average | 0.525 | 0.522 | **0.527** | 0.517 |
>
> On non-reasoning tasks, the performance gap between grown models and the baseline is relatively small, mirroring the observations on the Open-book and Closed-book Q&A benchmarks. On average, LIDAS only slightly outperforms the baseline and MIDAS, with LN-Scaling being the last.

---

> > ### Author Response · Authors · 2025-11-18
> >
> > ## Questions
> > 1. To ensure a strong and fair baseline, we use the exact same training setup as the original SmolLM-v1 family for all models (baseline, MIDAS, LIDAS, and LN-Scaling variants) to build upon established protocols without introducing confounding variables. We want to highlight that these protocols were likely tuned for the baseline, and we do not make any changes to them.
> > - Initialisation: For all layers, we use the standard parametrisation, initialising the bias with 0 and the weights with a normal distribution with a standard deviation of 0.022 for the 1.7B model and 0.042 for the 360M model. The “up projection” of the MLP, as well as the attention matrices, are scaled by $\frac{\sigma}{\sqrt{2 * nlayers}}$.
> > - Learning rate: For both models, we use a trapezoidal learning rate schedule with a linear warm-up of 2000 steps and a square root decay over the final 30000 steps. The peak rate of the 1.7B model is 0.0005, and for the 360M 0.003. Further details can be found in Appendix A.
> > - We do not apply any layer-wise LR scaling or depth-dependent learning-rate modifications of the kind proposed by Dey et al. [6], a paper which we note was published concurrently with the development of this paper. We agree that combining improved learning-rate scaling schemes with depth growth is an interesting direction for future work, potentially further improving results for the grown models [5]. Therefore, we added a brief discussion of Dey et al. [6] in the Related Work section of the revised manuscript.
> > 2. The results we observe in Figure 5 indicate that computation is not simply moved from earlier layers to later layers, but that computations are indeed differently distributed. Specifically, we observe in Figure 5a that skipping a layer in the middle, and especially at the beginning, has an effect on its downstream layers. Nevertheless, when confronting simpler tasks, we observe that some centre computations can be dropped with a slight decrease in performance (Figure 10).
> > 3. [2], who originally introduced MIDAS, motivate that this kind of symmetry is related to the behaviour of looped transformers, where repeated application of the same block has been linked to stronger reasoning abilities [4]. We do not claim that symmetry is universally optimal or should be explicitly enforced. Instead, we consider it an emergent structural property that correlates with the desirable behaviours of depth-grown models, which ultimately inspired the LIDAS modification. We clarified this interpretation and connected it better to looped transformers in the revision.
> > 4. We are not fully sure we understand the reviewer’s question and would like to clarify our interpretation below. Although regularly trained models can often be pruned more aggressively, we view this capacity not as an advantage, but as a consequence arising from structural redundancy inherent in conventional training, which one ultimately seeks to avoid. If pruning is desired, grown models allow for a more structured approach by pruning (i.e., skipping) layers or blocks in the middle, resulting in a more graceful performance degradation compared to the catastrophic drops of the baseline (Fig. 10).

---

> ### Comment · Reviewer_UGkv · 2025-11-19
> **Reviewer Response to Rebuttal**
>
> Weakness 1: This is good evidence. Thank you.
>
> Weakness 2: Thank you. With the addition of new results, it could be nice to add standard error to the tables for readers to distinguish significance of results, as 2ves also highlighted in their review and to me this answer is satisfactory. I think both
>
> Question 1: I agree Dey et al. should be classed as concurrent. These are all reasonable design choices. Thank you
>
> Question 2: Thank you.
>
> Question 3: Thank you.
>
> Question 4: Your interpretation is correct, your answer is also informative thank you.
>
> I think the addition of LN-scaling is a good one. Reviewer 2ves points out “Block-size may be an unexamined confound[er]” I agree with 2ves but also appreciate your response.
>
> Overall, I think this paper is a good step forward in this research area and recommend acceptance on that basis. I would urge authors to release the trained models to allow future research to build upon their efforts.

---

> > ### Author Response · Authors · 2025-11-24
> >
> > We thank the reviewer for their positive response to our rebuttal. We will include the additional seed experiments in the manuscript and release the trained models.

---

### Official Review · Reviewer_ZzFz · 2025-10-31

**Soundness:** 3
**Presentation:** 2
**Contribution:** 3
**Rating:** 2
**Confidence:** 3

**Summary:**

Transformers are known not to fully exploit the deeper layers, known as the Curse of Depth. Recent depth growing techniques adaptively increase the depth of a model and can obtain more gain from the depth. This study investigates the Curse of Depth through detailed analysis comparing non-growing and growing models. The detailed analysis reveals three main observations. (i) The removal of deeper layers has significantly greater impact for depth-grown models than non-grown ones, indicating the exploitation of deeper layers in depth-grown models. (ii) The depth-grown models are more robust against computational block permutations. (iii) Depth-grown models present cyclic patterns in the attention sublayer's contribution over layers.

**Strengths:**

- Extensive experiments are conducted on the impact of deeper layers, comparing standard depth-fixed models and recent depth-growing models.
- Besides, LIDAS, a variant of MIDAS method, is proposed, which attains superior performance in reasoning-intensive tasks.
- The presentation is easy to follow; the hypothesis, evidence, results, and interpretation are presented clearly.

**Weaknesses:**

This study experimentally collects observations on depth-non-growing and depth-growing models. While I appreciate them, one of the major weaknesses of this work is that the connection between these observations is unclear, and the practical takeaway from them is limited.

Depth-fixed models do not fully take advantage of the depth, and deeper layers can be dropped with a subtle cost in performance. This has been known already, and the experiments collect related observations from layer-wise analysis with a contrast to depth-growing models. The introduction [l.053] writes

> However, a clear mechanistic understanding of these gains has so far been missing.

but I don't feel this paper fully addresses these points. The experiments strengthen the known fact but do not offer how to boost the depth-growing models (if some understanding is obtained, this should be done to some extent, which validates the understanding).

The proposed method, LIDAS, is also kind of independent; it's not designed based on the observations. While LIDAS performs better than MIDAS in some tasks, the reason is not explained clearly or theoretically.

I'm afraid to say that this work is preliminary or a "concatenation" of several results without any conclusive remarks. Each of the experiments reports a solid observation, and I appreciate it. The main concern is the final takeaway (and its impact) built upon them in training Transformer models,

**Questions:**

Please address the weaknesses. Particularly, what new understanding of depth-fixed and depth-growing models is obtained from experiments, and how is it tested? Note that this is not asking about how the experiments to obtain new observations are performed (these are already well presented), but asking what new understanding/hypothesis is obtained from the observations, and how their correctness is justified.

---

> ### Author Response · Authors · 2025-11-18
>
> We thank the reviewer for the constructive comments on this work and for recognising its clarity of presentation (“the hypothesis, evidence, results, and interpretation are presented clearly”) and its scope of experiments (“Extensive experiments are conducted on the impact of deeper layers, comparing standard depth-fixed models and recent depth-growing models”). The experiments are designed to investigate why depth-grown models perform better in reasoning (through a more effective use of their depth), improve the understanding of their internal computations and link them to looped transformers, as motivated by MIDAS in [2]. We show that this connection arises because depth-grown models implement permutable, block-like computation along their depth. In the following sections, we will address all raised concerns in detail. For a high-level description of the main modifications, please consult the general response.
>
>
> **What new understanding of depth-fixed and depth-growing models is obtained from experiments**
>
> The three sets of experiments give complementary views on a single underlying picture of how depth-grown models organise computation in depth. As reviewer h9LG summarises nicely: “Converging diagnostics (tuned‑lens, early‑exit, swap/reverse/skip) make a coherent case that growth increases effective depth usage”. First, the layer-removal experiments demonstrate that depth-grown models genuinely rely on their later layers: removing deeper layers harms grown models significantly more than depth-fixed baselines, indicating that these layers perform essential, non-redundant computations rather than acting as unused capacity (Fig. 1). We then further investigate how this additional computation is arranged. The layer-swapping experiments (Fig. 3) reveal that grown models are sensitive to perturbations of very late layers but comparatively robust to block-wise interventions in the middle of the network, unlike depth-fixed baselines. This suggests that the central parts of a depth-grown model behave like permutable computational blocks. Finally, the cyclic pattern in attention contributions (Fig. 4) indicates that these blocks are not arbitrary: they share a recurring internal structure that repeats along depth, and reversing their internal order has a more detrimental impact on performance than other interventions (Fig. 6).
>
> Taken together, these experiments reveal that training by gradual growing in depth alters how Transformers utilise depth in a manner qualitatively distinct from depth-fixed training and closely aligned with Looped Transformers, a view proposed in [2] but not further explored. We highlighted these connections more clearly in the contributions mentioned in the introduction (at the end of Section 1) and the newly added discussion section (Section 5) in the revised manuscript.
>
>
> **LIDAS is not designed based on the observations.**
>
> We thank the reviewer for pointing this out and have taken it into account by revising the manuscript. Indeed, we have not designed LIDAS with respect to the observations on depth, but it is directly motivated by the observations on weight symmetry in MIDAS, summarised in Fig. 7a. [2] introduced MIDAS with the motivation that its growth scheme approximates the kind of (functional) symmetric structure observed in Looped Transformers. The analysis shows that, in practice, MIDAS does not exhibit a clean symmetric weight-similarity pattern because the growth is constrained by the block size. LIDAS is designed to restore this symmetry by growing at the layerwise middle, which yields a more symmetric weight-similarity pattern. We also want to emphasise that we do not present LIDAS as the main contribution of the paper. It is a small, targeted modification of MIDAS, inspired by the hypotheses regarding weight symmetry and the connection between depth-grown models and depth-recursive models. The analysis presented in this work, which we intend as the main contribution, was completed afterwards to mechanistically understand these depth-growth methods compared to standard non-grown training: Fig. 7a confirms that LIDAS yields more symmetric weights as intended, while the others provide insight on how MIDAS, LIDAS, and non-grown models behave “under the hood”.

---

> > ### Author Response · Authors · 2025-11-24
> >
> > We thank the reviewer again for the constructive feedback and suggestions. We have included the requested changes and clarifications (as well as additional experiments) in the updated manuscript and hope they address the main concerns:
> >
> > * **Clear statement of the main mechanistic insights**
> > * **Explicit links from the insight to specific experiments**
> > * **Clarified the motivation for the design of LIDAS**
> > * **Additional experiments to support the generality of our claims**
> >
> > We are happy to address any additional concerns, in particular any concrete suggestions for further experiments the reviewer might have.

---

### Official Review · Reviewer_h9LG · 2025-11-01

**Soundness:** 3
**Presentation:** 3
**Contribution:** 3
**Rating:** 4
**Confidence:** 2

**Summary:**

The paper studies depth‑growth training (e.g., MIDAS) as a remedy for the “curse of depth” observed in pre‑LN Transformers and proposes LIDAS, a layer‑wise middle‑duplication variant. Using SmolLM‑v1 at 360M and 1.7B parameters, the authors show that grown models (i) increase late‑layer utility according to multiple diagnostics (depth score, early‑exit overlap, tuned‑lens accuracy), (ii) exhibit permutable computational blocks in the middle of the network, and (iii) match or exceed MIDAS on reasoning while keeping NLL stable. Training compute is reported to decrease by ≈23% (1/1.29×) under the paper’s schedule.

**Strengths:**

- Mechanistic depth. Converging diagnostics (tuned‑lens, early‑exit, swap/reverse/skip) make a coherent case that growth increases effective depth usage.
- Actionable variant. LIDAS is a lightweight change that preserves or improves reasoning without harming NLL (i.e., token-level negative log-likelihood/perplexity on held-out text), indicating no regression in general language modeling quality).
- Reproducibility. Setups and intervention protocols are described clearly; the narrative is easy to follow.

**Weaknesses:**

- Compute accounting / fairness. Main comparisons fix steps, not FLOPs. Since growth changes training compute, a FLOPs‑matched baseline (e.g., truncating baseline steps to ≈77%) is needed to support “efficiency–performance” claims. Error bars (multi‑seed) are also missing on the headline numbers.
- Cross‑method context. The paper positions growth as a remedy for pre‑LN “curse of depth,” yet omits direct comparisons to LayerNorm scaling baselines. A small 2×2 factorial (Pre‑LN vs Mix‑LN) × (no growth vs LIDAS) would clarify whether growth is orthogonal or redundant with normalization tricks.
- Scale & breadth. Evidence is limited to 360M/1.7B and emphasizes reasoning; generalization to ≥7B/13B, instruction‑tuned, code, and knowledge‑heavy QA remains uncertain. Systems issues that may arise at scale (optimizer‑state copying, pipeline‑parallel rebalancing at growth boundaries) are not explored.

**Questions:**

- Backbone generality. How do findings transfer to post‑LN/parallel‑residual and MoE backbones?
- Curricula interplay. Are growth and length curriculum or token‑drop schedules complementary or redundant?
- Failure modes. Any tasks where growth hurts (knowledge‑heavy QA, code), and diagnostic clues as to why?
- Large‑scale stability. At ≥13B, do you observe optimizer‑state copy overheads or the need for short local warm‑ups at growth boundaries?

---

> ### Author Response · Authors · 2025-11-18
>
> We thank the reviewer for their detailed feedback and positive assessment, including that we “make a coherent case that growth increases effective depth usage”, “LIDAS is a lightweight change that preserves or improves reasoning without harming NLL”, and the “Setups and intervention protocols are described clearly”. We address the main weaknesses and questions below. Please refer to the general response for a summary of the main modifications in the revised manuscript.
>
> **Weaknesses**
>
> - **Compute accounting/fairness**: We thank the reviewer for this suggestion and agree that a FLOPs-matched baseline would strengthen the efficiency claims. We therefore ran a truncated baseline, as suggested, in which we reduced the number of training steps to approximately match the compute of MIDAS and LIDAS (see the general response). Note that, as expected, reducing the number of FLOPs for the baseline decreases its performance, thereby increasing the gap to the grown models in an iso-FLOP comparison. Due to the high computational cost, we, along with nearly all labs, are unable to run multi-seed experiments for the 1.7B model (especially within the time frame of the rebuttal). Instead, we added the results with another seed for the 360M model to the general response and verified that the findings still hold.
> - **Cross-method context**: We have updated the manuscript with new experiments using LayerNorm-Scaling [1] and a combination of LayerNorm-Scaling with LIDAS. We observe that LayerNorm-Scaling does not significantly improve benchmark performance over the baseline and is usually outperformed by LIDAS, especially at the 1.7B model size (Table 1). When combining LayerNorm-Scaling with LIDAS, we observe improvements over the LayerNorm-scaled baseline for the 360 model in all categories except Closed-book Q&A, as well as in reasoning-heavy tasks for the 1.7B size (Table 8). Additionally, growing clearly leads to better depth utilisation (Fig. 27, 28), indicating the benefits of depth-growth. In general, we find LayerNorm-Scaling to be ineffective at scale, and the experiments indicate that growth is orthogonal to normalisation tweaks.
> - **Scale & breadth**: We do acknowledge that validation at scale is an important factor. This also led us to run multiple models of size 1.7B, pre-trained on 400B tokens from scratch. Unfortunately, scaling to an even bigger model is beyond nearly all academic and many industrial labs due to computational reasons. We hope that the presented scales are considered to be sufficient to support the claim that our findings generalise across scales.

---

> > ### Author Response · Authors · 2025-11-18
> >
> > **Questions**
> >
> > - **Backbone generality**: We have focused the analysis on pre-LN Transformers precisely because they are known to underutilise depth [1,3] and are currently the most common Layer Norm variant. However, by combining growth with LayerNorm-Scaling, we observe that the main qualitative findings persist under a modified normalisation scheme (Fig. 27, 28), which we believe provides evidence that the results indeed hold for more general architectures.
> > - **Curricula interplay**: We would kindly ask the reviewer to clarify the goal of this experiment. While we do agree that these are possibly interesting future directions, we do not see a clear connection between these interventions and the central claims of this paper, which focus on how depth-grown models overcome the curse of depth. We would therefore consider such combinations as separate projects rather than as checks directly aligned with the current scope.
> > - **Failure Modes**: We observe that MIDAS can lead to worse performance compared to standard training on Closed Book benchmarks, i.e., particularly those designed to primarily extract knowledge from the model, and NLL. In contrast, LIDAS often matches the baseline even in those categories. **LIDAS matches or exceeds the baseline in all benchmark categories for the 1.7B model at 23% reduced FLOPs in Table 1.** While we cannot provide a definitive answer to this question, we hypothesise that this observation stems from the fact that grown models have a lower knowledge capacity, similar to looped models [4]. This aligns with [2], who motivate MIDAS by its similarity to looped models, which are known to improve reasoning but can decrease performance on knowledge-heavy tasks (e.g., [4]). Code and instruction tuning is a post-training technique that is out of scope due to limited compute resources. However, we provide results on math finetuning, which portrays results in this direction at more feasible scales.
> > - **Large-scale stability**: As discussed above, scaling to 13B would require significant computational resources and is not feasible, especially within the timeline of this rebuttal period. We do not believe that the optimiser-state copying would pose an overhead. This is a very inexpensive computation that occurs only a few times during training and can even be performed on CPU. No manual pipeline-parallel rebalancing is required at growth boundaries. After growing the model, we simply start a new run, and Nanotron (the library we use for pretraining) automatically re-partitions the model across devices.
> > Regarding the warm-ups close to growth boundaries: Since we copy optimiser and model weights, we start each grown layer from a stable initialisation. Therefore, at the scales we train, no adjustment of the learning rate is necessary.

---

> > > ### Author Response · Authors · 2025-11-24
> > >
> > > We thank the reviewer again for the constructive feedback and for suggesting additional experiments. We have included the requested experiments and clarifications in our rebuttal and hope they address the main concerns:
> > >
> > > * **FLOPs-matched baseline to support efficiency claims**
> > > * **Generality across architectures and scales**
> > > * **Scalability concerns**
> > > * **Failure modes of grown models**
> > > * **Comparisons to LayerNorm scaling baselines**.
> > >
> > > If any point would benefit from further detail, we’d be grateful to address any additional questions.

---

### Official Review · Reviewer_2ves · 2025-11-02

**Soundness:** 2
**Presentation:** 3
**Contribution:** 3
**Rating:** 6
**Confidence:** 3

**Summary:**

This paper provides a mechanistic investigation into why progressive layer growth improves downstream reasoning performance in Transformer LMs. Starting from MIDAS, the authors develop a layer-wise growth variant (LIDAS) and present a series of depth diagnostics (1) early-exit behavior, skip/swap/reverse perturbations, and block-wise contribution/similarity analyses and (2) showing that growth increases late-layer functional utilization and mitigates the Curse of Depth in pre-LN models.

## Soundness  2 (fair)
The analyses are methodologically sound, and the triangulation of probing, perturbation, and contribution metrics is compelling.
The primary limitation is that only 2 models of SmolLM family are studied, which is not sufficient to support the argument. Also, several claims are correlational, especially the interpretation that permutation robustness *implies* deeper utilization, which is plausible but not causally demonstrated.

## Presentation 3 (good)
The paper is exceptionally well written and well structured. The progression from motivation $\rightarrow$ analyses $\rightarrow$ implications is very clear. Some plots (Fig. 3, Fig. 4) assume background context that could be made more self-contained.

## Contribution - 3 (good)
The novelty lies in *mechanistic understanding*, not architecture: the work explains **why** depth growth improves performance, beyond reporting that it does. This deepens the conceptual grounding of growth-based reasoning improvements and is well suited for poster-level acceptance.

**Strengths:**

- Mechanistic insight into internal computation rather than only surface-level gains.
- Multiple convergent forms of evidence (early-exit, swap/reverse, contribution metrics).
- Clear link to Curse of Depth and demonstration of how growth reactivates late layers.
- Well-scoped and well-written with a strong explanatory narrative.
- Practical relevance for growth-based strategies in reasoning-oriented LMs.

**Weaknesses:**

### 1. Limited architectural generality
All experiments are on SmolLM (360M / 1.7B), a single pre-LN, short-context family. Since the contribution is mechanistic in nature, it remains unclear whether the observed “resurrected depth utilization” is a *property of staged growth itself*, or a *property of this architecture family*.

### 2. Causal link between permutation robustness and “depth utilization” remains implicit
For example, section 4.2 shows that grown models are more robust to block-level reordering, and **claims** this as evidence of deeper utilization. However, it needs more detailed explanations to bridge this casual relation.

### 3. Performance gaps are small and lack statistical framing
In Table 1, most LIDAS vs MIDAS gains are <1pp; without multi-seed variance this is difficult to interpret; if without confidence intervals or multi-seed reporting, the differences are too small to determine whether those small gains reflect a stable trend or training stochasticity. Since LIDAS is an architectural refinement, the size and stability of differences matter for how much weight the architecture (vs. the growth mechanism) contributes to the overall story.

### 4. The 360M + math-cooldown Primitive result is an unexplained outlier
In table 1, the improvement between LIDAS and MIDAS is dramatically larger, however, there's no explanation. (The paper itself only states that the cause of the improvements between grown model and baseline is unclear.)
This weakens the confidence that the measured gains derive cleanly from the proposed mechanism rather than an interaction or confound.

### 5. Block-size may be an unexamined confound
The periodicity observed in section 4.3 is shown only for block size b=4. If b=2 or b=8 produces a different cycle structure, part of the reported effect may be hyperparameter-induced. A small ablation would confirm whether the pattern is intrinsic or scheduler-driven.

### 6. Need more explanation on the causal relation between symmetry and depth depth utilization
Section 4.4 mentions that LIDAS is more block-wise symmetrical than MIDAS. However, it does not explain precisely why this offers an advantage in terms of in-depth utilization. Need more detailed explanation to build a causal link.
Also, better if there is a causal link between this symmetry and the mechanism of LIDAS.

**Questions:**

In Fig. 1 and Fig. 9, the difference between the two variants is subtle (In Fig.1, it seems that LIDAS is not as good as MIDAS). Is the benefit primarily symmetry/stability, or does it yield distinct behavior in downstream layer activation? Clarifying *where* LIDAS helps would sharpen the architectural takeaway.

Additionally, please address the weaknesses stated.

---

> ### Author Response · Authors · 2025-11-18
>
> We sincerely thank the reviewer for their interest and remarks. We appreciate the positive assessment of the paper’s scope, presentation and soundness (“The analyses are methodologically sound, and the triangulation of probing, perturbation, and contribution metrics is compelling”). We agree that several clarifications and small additions can strengthen the work. Below, we address the comments in detail; please see the general response for an overview of the main changes in the revised manuscript.
>
>
> **1. Limited architectural generality**
>
> We agree with the reviewer that demonstrating robustness across different architectures would help establish the generality of the presented mechanistic claims. However, the ability to perform analyses beyond one model family is inherently limited due to numerous reasons: (1) In order to achieve a fair comparison, we pretrain all models from scratch on fixed token budgets, which is a costly approach. Therefore, we chose to focus on a widely adopted Pre‑LN setup in SmolLM, which follows “Llama” model configurations and enables controlled analysis under a constrained budget. (2) We focus on the scaling axis, showing the generality of the claims across model scales.
> Running these expensive experiments on multiple architectures & scales would be challenging even for frontier level computational possibilities. To illustrate this point, published research works from leading industrial labs e.g. [2] restrict the analysis to one architectural family.
>
> To address this concern, however, we present additional results on models utilising LayerNorm-Scaling (LN-Scaling; [1]) as an architectural variant in Appendix E. We have trained both MIDAS and LIDAS with LN-Scaling and again find the same trends in depth utilisation and reasoning performance. This supports the claim that the reported findings are not an artefact of a single model instance but generalise across model size and variant architectures.
>
> **2. Causal link between permutation robustness and “depth utilization” remains implicit**
>
> The intent of Section 4.2 is not to use permutation robustness as proof of depth utilisation. Rather, the swap/reverse experiments show that growing changes depth-wise computations by altering the model's computational structure. We have revised the introduction and discussion of the manuscript to make this more explicit.
>
>
> **3. Performance gaps are small and lack statistical framing**
>
>  We appreciate this comment. For clarification, Table 1 reports aggregated scores across 21 benchmarks, grouped into Open-book Q&A, Closed-book Q&A, Language (Lambada & Hellaswag), MathWorld, and (Reasoning) Primitives. This aggregation already reduces variance, and the corresponding per-task breakdowns are provided in Appendix C. Moreover, the primary aim of this work is to provide a mechanistic understanding of depth-grown models rather than to maximise empirical gains alone.
> That said, we agree that explicitly accounting for uncertainty strengthens the conclusions. We therefore trained the 360M models with an additional random seed (see the general response, where we report these results in tabular form). The additional runs do not alter the main conclusion that LIDAS outperforms the baseline and MIDAS. Rather, they further support this pattern.

---

> > ### Author Response · Authors · 2025-11-18
> >
> > **4. The 360M + math-cooldown Primitive result is an unexplained outlier**
> >
> > The result of the 360M + math-cooldown Primitive experiment indeed appears as an outlier. To ensure full transparency, we provide a detailed per-primitive breakdown for the math-cooldown 360M models below.
> > | Model Name | Copying Random Words | Copying Real Words | Variable Assignment Basic | Variable Assignment Code | Variable Assignment Math |
> > |------------|--------------------------------|------------------------|----------------------------------|---------------------------------|---------------------------------|
> > | Baseline   | 17.3                           | 16.2                   | 25.2                             | 66.3                            | 40.6                            |
> > | MIDAS      | 16.1                           | 19.7                   | 20.2                             | 66.3                            | 53.4                            |
> > | LIDAS      | 50.4                           | 40.4                   | 30.2                             | 66.2                            | 64.6                            |
> >
> > From this decomposition, it becomes clear that the unusually high aggregated Primitive score is largely driven by the two copying tasks (primarily Copying Random Words, and secondarily Copying Real Words). We verified that all models were evaluated on exactly the same set of prompts, ruling out the possibility of a data or evaluation bug. Moreover, in the additional-seed results table (see general response), the performance gap is reduced. However, LIDAS still maintains a clear advantage over the baseline. Finally, we emphasise that finetuning, here, math-focused cooldown training, can amplify performance differences between standard baselines and grown models, a phenomenon also observed in prior work [2].
> >
> > **5. Block-size may be an unexamined confound**
> >
> > We agree that ablating the block size is important for assessing the generalizability of the reported claims about cyclical patterns. To this end, we have added experiments with block size 8 for the 360M model (Fig. 23) and block size 3 for the 1.7B model (Fig. 24). These results confirm that the emergence and structure of the cyclical pattern are indeed block-size dependent. For example, the maximum (across block layers) of the Mean Relative Contribution and the Cosine Similarity appear at the last layer of each block (similar to the block size 4 model); see Appendix D.3 for further details.
> >
> > At the same time, block size also affects downstream performance. In the experiments, a smaller block size of 3 yielded slight performance improvements, whereas a larger block size of 8 led to mild degradation.
> >
> > **6. Need more explanation on the causal relation between symmetry and depth utilization**
> >
> > We do not claim that this enhanced weight symmetry directly causes higher depth usage. Instead, we investigate the symmetry to support the motivation for LIDAS. LIDAS duplicates layers exactly at the model’s centre, whereas MIDAS duplicates a block that is only near the centre. This design choice brings LIDAS closer to the original motivation of MIDAS [2], yielding more symmetric weights that better mirror the symmetric functional similarities of Looped Transformers (e.g., ALBERT in [2]).
> >
> > **MIDAS vs. LIDAS**
> >
> > Identifying a single dominant factor that gives LIDAS an advantage in downstream tasks is not straightforward, and indeed, Figs. 1 and 9 do not by themselves isolate a unique mechanism. However, based on the current analysis, we see two qualitative differences that most clearly distinguish LIDAS from MIDAS:
> > 1. Higher weight similarity across blocks. LIDAS exhibits larger inter-block weight similarity (as a direct consequence of its growing strategy), bringing it closer in spirit to looped architectures.
> >
> >
> > 2. Stronger early attention contributions in mid-blocks. LIDAS consistently enhances both the contribution and the alignment of the first attention sublayer within each mid-block to the residual stream (see Figs. 4 and 7b), suggesting a more effective use of these layers for downstream computation.
> >
> > For a qualitative and quantitative discussion of LIDAS vs. MIDAS, please refer to section 5.

---

> > > ### Author Response · Authors · 2025-11-24
> > >
> > > We thank the reviewer again for the constructive reviews and for suggesting additional experiments. We have included the requested experiments and clarifications in our rebuttal and hope they address the main concerns:
> > >
> > > - **Generality across architectures and scales**
> > > - **Link between permutation robustness and depth utilisation**
> > > - **Statistical framing & robustness**
> > > - **Clarify Outlier 360M + math‑cooldown**
> > > - **Examine Block-Size as a confound**
> > > - **Relation of symmetry and depth utilisation**
> > > - **MIDAS vs. LIDAS**
> > >
> > > If any point would benefit from further detail, we’d be grateful to address any additional questions.

---

### Author Response · Authors · 2025-11-18
**General Response**

We thank all reviewers for their careful reading and constructive feedback. In particular, we are grateful to read that this work “deepens the conceptual grounding of growth-based reasoning improvements and is well suited for poster-level acceptance” (Reviewer 2ves), proposing a “lightweight change that preserves or improves reasoning without harming NLL” (Reviewer h9LG), “attains superior performance in reasoning-intensive tasks” (Reviewer ZzFz), and “the analysis presented is deep and thorough” (Reviewer UGkv). All reviewers also noted that this work illuminates and addresses the “Curse of Depth” in Transformers.

In response to recurring concerns, we have conducted additional experiments and updated the manuscript accordingly. Changes are highlighted in green, with green titles indicating an entirely new section. We summarise the main modifications below:

**Baseline with LayerNorm-Scaling**

We have added an additional baseline trained with the recently proposed LayerNorm-Scaling scheme [1], which is designed to control exploding variance in later layers of Pre-LN architectures and has been reported to outperform Pre-LN, Post-LN, and Mix-LN variants in task performance while increasing depth usage. With the current setup, however, this baseline does not improve depth utilisation and yields only small, incremental gains in benchmark performance over the original baseline. Moreover, these gains largely vanish when scaling to the larger 1.7B model (Table 1; Appendix E).

**LIDAS + LayerNorm-Scaling**

We further introduce a variant that combines LayerNorm-Scaling with LIDAS (Table 8; Appendix E). This model achieves improved reasoning benchmark performance over the corresponding LayerNorm-Scaling baseline (without growth) and exhibits stronger depth utilisation (Fig. 27, 28). These results indicate that gradual depth growth remains beneficial and is largely orthogonal to the choice of normalisation scheme.

**FLOPs-matched comparisons**

To enable a more direct efficiency comparison between growth and non-growth training, we trained truncated baselines for both the 360M and 1.7B models to match the FLOPs of the grown models. This FLOPs-matched setup strengthens the reported efficiency claims.
| Scale | Model              | NLL ↓    | Open-book Q&A (F1 ↑) | Closed-book Q&A (F1 ↑) | Lambada (Acc ↑) | Hellaswag (Acc ↑) | MathWorld (Acc ↑) | Primitives (Acc ↑) | MathWorld Math Cooldown (Acc ↑) | Primitives Math Cooldown (Acc ↑) |
|-------|--------------------|----------|-----------------------|------------------------|-----------------|-------------------|------------------------|-------------------------|-------------------------|--------------------------|
| 360M  | Baseline Truncated | 2.19     | 20.72                 | 14.17                  | 42.65           | 39.21             | 3.07                   | **32.98**               | 7.30                    | 36.90                    |
| 360M  | MIDAS              | 2.18     | 24.57                 | 13.75                  | 43.31           | 40.36             | **4.39**               | 28.18                   | **13.43**               | 35.14                    |
| 360M  | LIDAS              | **2.16** | **26.63**             | **14.57**              | **44.03**       | **40.58**         | 4.36                   | 31.20                   | 12.30                   | **50.36**                |
| 1.7B  | Baseline Truncated | 1.97     | 29.03                 | 17.99                  | 50.22           | 45.60             | 12.36                  | 31.04                   | 21.15                   | 41.20                    |
| 1.7B  | MIDAS              | 1.97     | 28.80                 | 18.50                  | 50.81           | 46.19             | 16.07                  | 40.88                   | 24.01                   | **55.46**                |
| 1.7B  | LIDAS              | **1.96** | **29.84**             | **19.08**              | **51.41**       | **46.32**         | **18.59**              | **47.34**               | **24.60**               | 53.00                    |

---

> ### Author Response · Authors · 2025-11-18
> **General Response - continued**
>
> **Multi-seed runs at 360M**
>
> We trained other instances of the 360m model from scratch, using a different seed. In general, the superiority of LIDAS over the baseline and MIDAS persists across these additional runs, indicating that the observed gains follow a stable pattern rather than arising from training stochasticity.
>
> | Model    | NLL ↓   | Open-book Q&A (F1 ↑) | Closed-book Q&A (F1 ↑) | Lambada (Acc ↑) | Hellaswag (Acc ↑) | MathWorld (Acc ↑) | Primitives (Acc ↑) | MathWorld Math Cooldown (Acc ↑) | Primitives Math Cooldown (Acc ↑) |
> |----------|---------|----------------------|------------------------|-----------------|-------------------|------------------------|-------------------------|-------------------------|--------------------------|
> | Baseline | 2.18    | 23.18                | 14.22                  | 43.16           | 40.16             | 3.11                   | 29.92                   | 7.91                    | 37.36                    |
> | Midas    | 2.18    | 24.26                | **14.34**              | 42.58           | 40.11             | **3.47**               | 34.08                   | 8.50                    | 41.86                    |
> | Lidas    | **2.16**| **25.02**            | 14.08                  | **44.27**       | **40.90**         | 2.59                   | **37.14**               | **10.47**               | **46.88**                |
>
> **Conclusion**
>
> We have also added a dedicated discussion section that synthesises these new experiments with the original analyses, making the overall narrative and empirical evidence more explicitly connected.
>
> We have tried our best to address the reviewers' concerns in the responses. We look forward to receiving additional feedback and eagerly anticipate any further discussions to enhance the clarity and scope of the work.
>
> References.
>
> [1] The Curse of Depth in Large Language Models, Wenfang Sun and Xinyuan Song and Pengxiang Li and Lu Yin and Yefeng Zheng and Shiwei Liu, 2025
>
> [2] On the inductive bias of stacking towards improving reasoning, Nikunj Saunshi, Stefani Karp, Shankar Krishnan, Sobhan Miryoosefi, Sashank Jakkam Reddi, and Sanjiv Kumar, 2025
>
> [3] Do language models use their depth efficiently? Robert Csordas, Christopher D Manning, and Christopher Potts, 2025
>
> [4] Reasoning with Latent Thoughts: On the Power of Looped Transformers, Nikunj Saunshi and Nishanth Dikkala and Zhiyuan Li and Sanjiv Kumar and Sashank J. Reddi, 2025
>
> [5] Efficient Construction of Model Family through Progressive Training Using Model Expansion, Kazuki Yano and Sho Takase and Sosuke Kobayashi and Shun Kiyono and Jun Suzuki, 2025
>
> [6] Don't be lazy: CompleteP enables compute-efficient deep transformers, Nolan Dey and Bin Claire Zhang and Lorenzo Noci and Mufan Li and Blake Bordelon and Shane Bergsma and Cengiz Pehlevan and Boris Hanin and Joel Hestness, 2025

---

### Meta-Review · Area_Chair_KfzS · 2025-12-12

**Summary:**

The paper present an empirical investigation of depth-grown models. Reviewers appreciated the content and clarity of the paper, but were nonetheless critical of the experimental scope and, to a lesser extent, of whether the experimental findings are weaved into a compelling conclusion. The authors wrote very extensive rebuttals with a non-trivial amount of new content and results that may by themselves surpass the scope of rebuttal revisions, particularly in the unique circumstances of the present conference where no further reviewer engagement is possible. The recommendation is therefore to decline the paper and encourage a resubmission.

**Reviewer Concerns:**

Concerns about experimental scope were acknowledged and explained by the high cost of larger-scale experiments for the topic in question. This is undoubtedly true and understandable, but is also a consideration the reviewers had been almost certainly aware of while writing their reviews, and therefore presenting it in the rebuttal would have been unlikely to change their minds. New experimental results and explanations might have prompted reviewers to moderately revise their scores upward but this is unlikely to have changed the final outcome.

**Reviewer Scores:**

While the rebuttal was extensive and contributed new content, ultimately there was little in the way of misunderstandings that would have led to a dramatic shift in scores. My best speculation of how a reviewer discussion would have unfolded is that some of the three reviewers who originally recommended a reject would have revised their score upward by a notch as a show of appreciation for the authors' contributions in the rebuttal, but the overall picture would still be of a borderline paper just under the bar that would have been recommended for rejection.

---

### Decision · Program_Chairs · 2026-01-26

Reject